# Tradeoffs between Mistakes and ERM Oracle Calls in Online and Transductive Online Learning

Idan Attias [*]        Steve Hanneke[†]        Arvind Ramaswami[‡]

## Abstract

We study online and transductive online learning in settings where the learner can interact with the concept class only via Empirical Risk Minimization (ERM) or weak consistency oracles on arbitrary subsets of the instance domain. This contrasts with standard online models, where the learner has full knowledge of the concept class. The ERM oracle returns a hypothesis that minimizes the loss on a given subset, while the weak consistency oracle returns only a binary signal indicating whether the subset is realizable by a concept in the class. The learner's performance is measured by the number of mistakes and oracle calls.

In the standard online setting with ERM access, we establish tight lower bounds in both the realizable and agnostic cases: $\Omega(2^{d_{\mathrm{LD}}})$ mistakes and $\Omega(\sqrt{T 2^{d_{\mathrm{LD}}}})$ regret, respectively, where $T$ is the number of timesteps and $d_{\mathrm{LD}}$ is the Littlestone dimension of the class. We further show how existing results for online learning with ERM access translate to the setting with a weak consistency oracle, at the cost of increasing the number of oracle calls by $O(T)$.

We then consider the transductive online model, where the instance sequence is known in advance but labels are revealed sequentially. For general Littlestone classes, we show that the optimal mistake bound in the realizable case and in the agnostic case can be achieved using $O(T^{d_{\mathrm{VC}}+1})$ weak consistency oracle calls, where $d_{\mathrm{VC}}$ is the VC dimension of the class. On the negative side, we show that $\Omega(T)$ weak consistency queries are necessary for transductive online learnability, and that $\Omega(T)$ ERM queries are necessary to avoid exponential dependence on the Littlestone dimension. Finally, for special families of concept classes, we demonstrate how to reduce the number of oracle calls using randomized algorithms while maintaining similar mistake bounds. In particular, for Thresholds on an unknown ordering, $O(\log T)$ ERM queries suffice, and for $k$-Intervals, $O(T^3 2^{2k})$ weak consistency queries suffice.

## 1   Introduction

Online learning is a fundamental sequential prediction framework in which a learner receives a stream of instances and must predict their labels one at a time, observing the true label after each prediction [CBL06, SS+12, MRT12, SSBD14]. The goal is to minimize the number of mistakes compared to the best concept in a fixed concept class. A closely related framework is transductive online learning, where the entire sequence of instances is revealed in advance, removing instance uncertainty and isolating the challenge of predicting the labels [BDKM97, KK05, CBS13, HMS23, HRSS24]. This intermediate model retains the sequential nature of online learning while allowing the learner to prepare for the specific instances it will encounter.

---

[*]Institute for Data, Econometrics, Algorithms, and Learning (IDEAL), hosted by UIC and TTIC; idanattias88@gmail.com.

[†]Department of Computer Science, Purdue University; steve.hanneke@gmail.com.

[‡]Department of Computer Science, Purdue University; ramaswa4@purdue.edu.

39th Conference on Neural Information Processing Systems (NeurIPS 2025).

Both the online and transductive online settings have been studied extensively. The seminal work of Littlestone [Lit88] characterizes the optimal mistake bound attainable in the online binary classification setting in the realizable case, that is, when there exists a concept in the class that labels all instances correctly. This bound is expressed in terms of a complexity measure of the concept class $\mathcal{C}$ known as the Littlestone dimension (denoted by $d_{\mathrm{LD}}$). In particular, the optimal mistake bound is $d_{\mathrm{LD}}(\mathcal{C})$ and is achieved by the Standard Optimal Algorithm (SOA). In the agnostic setting, where labels may be arbitrary and the learner aims to minimize regret (the difference between the learner's cumulative mistakes and that of the best concept in the class), Ben-David et al. [BDPSS09] and Alon et al. [ABED$^{+}$21] showed that the Littlestone dimension also characterizes the optimal regret. These results also hold trivially in the transductive online setting, since the learner has more information than in the standard online setting. However, since the instances are known in advance, a sequence-dependent bound of $O\left(d_{\mathrm{VC}}(\mathcal{C})\log T\right)$ on the number of mistakes can be obtained [KK05] by counting the number of shattered sets and applying the Halving algorithm, where $d_{\mathrm{VC}}$ is the VC dimension [VC71, VC$^{+}$74] of the class and $T$ is the length of the instance stream. Most recently, it has been shown that for some classes, a mistake bound of $O(\sqrt{d_{\mathrm{LD}}})$ is achievable in the transductive setting, and that this rate is optimal [CHMS25].

While the SOA achieves the optimal mistake bound and has been used as an algorithmic primitive in many settings, implementing it efficiently poses significant computational challenges. The algorithm involves computing the Littlestone dimension multiple times during its online interaction, but even approximating the Littlestone dimension within any constant factor is computationally intractable [FL98, MR17, Man22]. Additionally, Hasrati and Ben-David [HBD23] showed that there exist recursively enumerable representable classes with finite Littlestone dimension that admit no computable SOA. As a result, Assos et al. [AAD$^{+}$23] and Kozachinskiy and Steifer [KS24] proposed a more realistic computational model for online learning based on oracle access to Empirical Risk Minimization (ERM). Given a labeled dataset, the ERM oracle returns a concept in the class that minimizes the error on the dataset. This oracle performs a simple and well-studied task, known to be sufficient in stochastic batch settings such as PAC learning. The central question, then, is: what is the mistake bound (or regret) when the online learning algorithm is restricted to using only the ERM oracle? A mistake bound that is exponential in the Littlestone dimension is attainable, as shown by [AAD$^{+}$23, KS24]. At the same time, an exponential mistake bound is also unavoidable [KS24] when the algorithm has access only to a "restricted" ERM oracle, one that can query only instance-label pairs generated by the adversary.

The goal of this paper is to study the power and limitations of various oracles in the online and transductive online learning settings. Crucially, unlike in standard settings where the learner has full access to the concept class, here the learner interacts with the class only through an oracle. The performance of the learning algorithm is measured by two parameters: the number of mistakes (or regret in the agnostic setting) and the number of oracle queries made, with the goal of achieving efficient oracle complexity, where only polynomially many oracle calls are made.

We begin by considering a general ERM oracle that, given any finite subset of the instance domain along with any labeling, returns an empirical risk minimizer. This contrasts with the more restricted ERM oracle used in prior work [AAD$^{+}$23, KS24], which allows queries only on instances and labels provided by the adversary during the interaction. For this more powerful oracle, we investigate two main questions:

- *In online learning, can we circumvent the exponential dependence on the Littlestone dimension in the mistake bound using the general ERM oracle?*

- *In the transductive online setting, where the learner has access to the full sequence of instances in advance, can this additional information lead to improved learning guarantees?*

Additionally, we consider a weaker oracle, the weak consistency oracle, which, given a labeled dataset, returns only a binary signal indicating whether the dataset is realizable by some concept in the class. This oracle can be viewed as solving the decision problem of realizability, in contrast to the ERM oracle, which solves the corresponding optimization problem. Surprisingly, Daskalakis and Golowich [DG24] recently showed that access to such an oracle can be used to construct a randomized PAC learning algorithm for binary and multiclass classification, regression, and learning partial concepts, with only a mild blowup in sample complexity and a polynomial number of oracle calls. In this paper, we explore the role of this oracle in online learning:

- *Can we construct algorithms for online and transductive online learning that make only a polynomial number of calls to the weak consistency oracle?*

For a comprehensive literature review, see Appendix A.

## 1.1 Our Contribution

We first study online learning with access to an ERM oracle that can query any subset of the domain, in contrast to the "restricted ERM" studied by [AAD+23, KS24], which can query only pairs of instances and labels generated by the adversary throughout the interaction. We then consider learning with a weak consistency oracle, recently introduced by [DG24] in the context of PAC learning.

**Results for Online Learning (Section 3).**  See a summary of the results in Table 1.

- We first prove a lower bound of $\Omega(2^{d_{\mathrm{LD}}})$ mistakes in the realizable setting (where $d_{\mathrm{LD}}$ is the Littlestone dimension), which holds for any learning algorithm that makes a finite number of ERM oracle calls. While [KS24] proved a stronger lower bound of $\Omega(3^{d_{\mathrm{LD}}})$, their result applies to the restricted ERM. In contrast, our lower bound holds for the general ERM and introduces several new challenges. We also prove a lower bound of $\Omega(\sqrt{T2^{d_{\mathrm{LD}}}})$ on the regret in the agnostic setting, which is the first lower bound of its kind. These two lower bounds match the known upper bounds of [AAD+23, KS24] in their exponential dependence on the Littlestone dimension (differing only by a constant factor in the exponent). This demonstrates that even with access to the general ERM, the mistake bound grows exponentially with the Littlestone dimension, unlike in standard online learning, where the learner has full access to the concept class and the mistake bound is $d_{\mathrm{LD}}$.

- We show that any deterministic online learning algorithm using the restricted ERM can be simulated using the weak consistency oracle, at the cost of increasing the number of oracle calls by $O(T)$. Consequently, the mistake bounds in existing results [AAD+23, KS24] can be achieved with an $O(T)$ blow-up when using the weak consistency oracle.

- We show a negative result for online learning of partial concept classes (concepts that may be undefined on certain parts of the domain) with access to an ERM oracle. Specifically, we construct a family of partial concept classes with $d_{\mathrm{LD}} = 1$ for which any algorithm must make $\Omega(T)$ mistakes. This stands in sharp contrast to the offline PAC setting, where learning partial concepts with a weak consistency oracle is feasible [DG24].

| ONLINE LEARNING: ORACLE COMPLEXITY–REGRET TRADEOFFS FOR LITTLESTONE CLASSES | | | | | |
|---|---|---|---|---|---|
| **Deterministic/Randomized Algorithm** | **Realizability** | **Oracle Type** | **Oracle Calls** | **Regret/Mistakes** | **Reference** |
| Deterministic | Realizable | Restricted ERM | $2^{O(d_{\mathrm{LD}})}$ | $2^{O(d_{\mathrm{LD}})}$ | [AAD+23, KS24] |
| Deterministic | | Weak Consistency | $2^{O(d_{\mathrm{LD}})}T$ | $2^{O(d_{\mathrm{LD}})}$ | Theorem 3.3 |
| Deterministic | | Restricted ERM | Finite | $\Omega(3^{d_{\mathrm{LD}}})$ | [KS24] |
| Randomized | | Agnostic ERM | Finite | $\Omega(2^{d_{\mathrm{LD}}})$ | Theorem 3.1 |
| Deterministic | Agnostic | Agnostic ERM | $T^{2^{O(d_{\mathrm{LD}})}}$ | $\tilde{O}\left(\sqrt{T2^{O(d_{\mathrm{LD}})}}\right)$ | [AAD+23] |
| Randomized | | Agnostic ERM | Finite | $\Omega(\sqrt{T2^{d_{\mathrm{LD}}}})$ | Theorem 3.2 |

Table 1: The learning model and oracles are defined in Section 2. The weak consistency oracle outputs whether a labeled sequence is realizable. We define three ERM oracles, ordered from weakest to strongest: the restricted ERM oracle, the ERM oracle, and the agnostic ERM oracle. $T$ is the number of timesteps, $d_{\mathrm{LD}}$ is the Littlestone dimension and $d_{\mathrm{VC}}$ is the VC dimension of the underlying concept class.

Given the negative results in the online setting, particularly the exponential growth in the number of mistakes, we consider the transductive online learning model, in which the instance sequence is known at the beginning of the interaction, but the labels are revealed sequentially by the adversary.

**Results for Transductive Online Learning (Section 4).**  See a summary of the results in Table 2. We start with results for general concept classes.

- We show how to discover all labelings of the concept class on a given set of instances using $O(T^{d_{\mathrm{VC}}+1})$ weak consistency oracle calls. With this preprocessing step, we can recover the optimal mistake bound and regret, which are currently known to be upper bounded by $O\left(\min\{d_{\mathrm{LD}}, d_{\mathrm{VC}}\log T\}\right)$ and $\tilde{O}\left(\sqrt{T\min\{d_{\mathrm{LD}}, d_{\mathrm{VC}}\log T\}}\right)$, respectively. This shows that the exponential dependence on the Littlestone dimension can be circumvented in the transductive setting using only a polynomial number of weak consistency oracle calls.

- On the other hand, we establish the following two lower bounds. We show that limiting the learner to $\Omega(T)$ weak consistency queries is necessary for transductive online learnability, and that restricting the learner to $\Omega(T)$ ERM queries is necessary to avoid exponential dependence on the Littlestone dimension.

We proceed to studying special families of concept classes, showing improved results, which highlight that randomization seems to be crucial for the improvements.

- Thresholds: We prove various upper bounds for deterministic and randomized algorithms using both oracles. The main result is a randomized algorithm with the ERM oracle that achieves an $O(\log T)$ mistake bound and $O(\log T)$ expected queries on the class of thresholds with unknown ordering. This represents an exponential improvement over deterministic algorithms.

- $k$-Intervals: We show there exists a randomized algorithm with the weak consistency oracle that achieves an optimal mistake bound (upper bounded by $O(k\log T)$) with $O(T^3 2^{2k})$ expected number of queries.

- $d$-Hamming Balls: For the ERM oracle with $d$-Hamming balls, we show a mistake bound of $2d$ using a single query. We also show a mistake bound of $d$ with $2^{d+1}$ queries.

In Section 5 we pose the main open problems for future work.

| | | TRANSDUCTIVE ONLINE LEARNING: ORACLE COMPLEXITY–REGRET TRADEOFFS | | | | |
|---|---|---|---|---|---|---|
| **Concept Class** | **Det. / Rand. Algorithm** | **Realizability** | **Oracle Type** | **Oracle Calls** | **Regret/Mistakes** | **Reference** |
| Littlestone Classes | Deterministic | Realizable | ERM | $2^{O(d_{\mathrm{LD}})}$ | $2^{O(d_{\mathrm{LD}})}$ | [AAD$^+$23, KS24] |
| | Deterministic | Realizable/Agnostic | Weak Consistency | $O(T^{d_{\mathrm{VC}}+1})$ | Optimal | Theorem 4.2 |
| | Deterministic | Realizable | ERM | $O(T)$ | $\Omega(2^{d_{\mathrm{LD}}})$ | Theorem 4.4 |
| | Randomized | Realizable | Weak Consistency | $O(T)$ | $\Omega(T)$ | Theorem 4.3 |
| Thresholds | Deterministic | Realizable | ERM | $O(T)$ | $O(\log T)$ | Theorem 4.5 |
| | Randomized | Realizable | ERM | $O(\log T)$ | $O(\log T)$ | |
| | Deterministic | Realizable | Weak Consistency | $O(T\log T)$ | $O(\log T)$ | |
| | Randomized | Realizable | Weak Consistency | $O(T)$ | $O(\log T)$ | |
| $k$-Intervals | Randomized | Realizable | Weak Consistency | $O(T^3 2^{2k})$ | $O(k\log(T))$ | Theorem 4.6 |
| $d$-Hamming Balls | Deterministic | Realizable | ERM | $O(1)$ | $O(d)$ | Theorem H.4 |
| | Deterministic | Realizable | Weak Consistency | $O(T)$ | $O(d)$ | |

Table 2: The lower bound for the weak consistency oracle from Theorem 4.3 applies to all three families of concept classes: thresholds, $k$-intervals, and $d$-Hamming balls, and is stronger than the bound for arbitrary Littlestone classes. We use the term "optimal" to indicate that the mistake bounds match those from standard transductive learning (where the concept class is known). In particular, the best known bounds depend linearly on the Littlestone or VC dimensions.

## 2    The Learning Models: Online and Transductive Online Learning with Oracle Access

We start with a definition of the online learning model where the interaction of the learner with the concept class is done only through an oracle. This is in contrast to the standard online model, where the learner knows the concept class in advance. The learning protocol is a sequential game between a learner and an adversary. Let $\mathcal{C} \subset \{0, 1\}^{\mathcal{X}}$ be a concept class, where $\mathcal{X}$ is the instance space and $\{0, 1\}$ is the label space. Let $\mathcal{F}$ be a family of concept classes, known to the learner (e.g., classes

with finite Littlestone dimension $d_{LD}$), such that $C$ is an unknown concept class from $\mathcal{F}$ chosen by the adversary. Suppose the learner only has oracle access to $C$ via an oracle $\mathcal{O}$. The sequential game proceeds for $T$ rounds, as follows. For each $t \in [T]$:

1. The adversary chooses $(x_t, y_t) \in \mathcal{X} \times \{0, 1\}$.
2. The learner observes $x_t$, picks a distribution $\Delta_t \in \Delta(\{0, 1\})$, and predicts $\hat{y}_t \sim \Delta_t$.
3. The adversary reveals $y_t \in \{0, 1\}$ and the learner suffers a loss $\mathbb{I}[\hat{y}_t \neq y_t]$.

We define $x_{1:T} = (x_1, x_2, \ldots, x_T) \in \mathcal{X}^T$ as the sequence of instances and $y_{1:T} = (y_1, y_2, \ldots, y_T) \in \{0, 1\}^T$ as the sequence of corresponding labels chosen by the adversary. In the realizable setting, the adversary is subject to the constraint that the sequence $(x_{1:T}, y_{1:T})$ is realizable by $C$, meaning that there exists $c \in C$ satisfying $c(x_i) = y_i$ for all $i \in [T]$. The performance of a learning algorithm $\mathcal{A}$ is measured by two metrics, the number of mistakes (or regret, in the agnostic setting) and the number of oracle queries. The mistakes are defined as follows:

$$M(\mathcal{A}, \mathcal{O}(C), x_{1:T}, c) = \sum_{t=1}^{T} \mathbb{I}[\hat{y}_t \neq c(x_t)].$$

The worst case number of mistakes of a learning algorithm $\mathcal{A}$ is defined as

$$M(\mathcal{A}, T) = \sup_{C \in \mathcal{F}} \sup_{c \in C} \sup_{x_{1:T} \in \mathcal{X}^T} \mathbb{E}[M(\mathcal{A}, \mathcal{O}(C), x_{1:T}, c)].$$

Similarly, $Q(\mathcal{A}, \mathcal{O}(C), x_{1:T}, c)$ is the total number of queries, defined as

$$Q(\mathcal{A}, T) = \sup_{C \in \mathcal{F}} \sup_{c \in C} \sup_{x_{1:T} \in \mathcal{X}^T} \mathbb{E}[Q(\mathcal{A}, \mathcal{O}(C), x_{1:T}, c)].$$

Here, the expectation is over the randomness of the algorithm.

We assume that the adversary is oblivious with respect to the choice of concept class $C$, target concept $c$, and instance sequence $x$, meaning that these are fixed in advance and do not depend on the learner's predictions or queries. Note that for deterministic algorithms, an oblivious adversary is as powerful as an adaptive adversary.

In the agnostic case, the sequence $(x_{1:T}, y_{1:T})$ is no longer constrained to be realizable by $C$, and the measure of performance is the regret, defined as

$$\text{Reg}(\mathcal{A}, \mathcal{O}(C), (x_{1:T}, y_{1:T})) = \sum_{t=1}^{T} \mathbb{I}[\hat{y}_t \neq y_t] - \inf_{c \in C} \sum_{t=1}^{T} \mathbb{I}[c(x_t) \neq y_t]$$

Define the worst case regret of the algorithm $\mathcal{A}$ as

$$\text{Reg}(\mathcal{A}, T) = \sup_{C \in \mathcal{F}} \sup_{(x,y) \in (\mathcal{X} \times \{0,1\})^T} \mathbb{E}[\text{Reg}(\mathcal{A}, \mathcal{O}(C), (x, y))],$$

and $Q(\mathcal{A}, T)$ in the agnostic case is defined as the worst-case total number of queries, over $C \in \mathcal{F}$ and $(x, y) \in (\mathcal{X} \times \{0, 1\})^T$.

Crucially, the learner knows the family of concept classes $\mathcal{F}$ but not $C \in \mathcal{F}$. The learner's only access to the concept class $C$ is through oracles defined below. We start with a few variants of the ERM oracle.

**Definition 2.1 (Variants of ERM Oracle)** We define the oracles in order of expressive strength. Let $C$ be the concept class we have access to.

- "Restricted ERM": At round $t$, given a subsequence $S$ of the pairs generated by the adversary, $(x_{1:t-1}, y_{1:t-1})$, if $S$ is realizable by $C$, then the ERM oracle returns some $c \in C$ consistent with $S$. Otherwise, it returns "not realizable." This oracle was used for the upper bounds in [AAD+23, KS24] and for the lower bound in [KS24]. Here, it is used primarily for comparison to our results, as well as in the upper bound in Theorem 3.3.

- ERM: Given any subset $S \subset \mathcal{X} \times \{0, 1\}$, if $S$ is realizable by $C$, then the ERM oracle returns some $c \in C$ consistent with $S$. Otherwise, it returns "not realizable". This oracle can be thought of as a "realizable" ERM or consistency oracle. For simplicity, we refer to it as ERM. This is the main variant used in this paper.

- "Agnostic" ERM: Given any subset $S \subset \mathcal{X} \times \{0, 1\}$, returns a a concept with the minimal error: $\operatorname{argmin}_{c \in \mathcal{C}} \sum_{(x,y) \in S} \mathbb{I}[c(x) \neq y]$. This oracle is used for the lower bounds in Theorems 3.1 and 3.2.

Additionally, we consider the following weaker oracle, studied by [DG24] in the context of PAC learning.

**Definition 2.2 (Weak Consistency Oracle)** Given a concept class $\mathcal{C} \subseteq \{0, 1\}^{\mathcal{X}}$ and any sequence $S = (x_1, y_1), ..., (x_n, y_n)$, the Weak Consistency Oracle returns "realizable" if there exists some $c \in \mathcal{C}$ consistent with $S$. Otherwise, it returns "not realizable".

We also study the transductive online setting with oracle access, where the only difference is that the adversary first selects a sequence $x = (x_1, x_2, \ldots, x_T) \in \mathcal{X}^T$, which is revealed to the learner in advance [4]. Then, the sequential interaction begins, with the adversary revealing the labels $y_t$ one by one. Formally, for each $t \in [T]$ :

1. The adversary chooses $y_t \in \{0, 1\}$.
2. The learner picks a distribution $\Delta_t \in \Delta(\{0, 1\})$, and predicts $\hat{y}_t \sim \Delta_t$.
3. The adversary reveals $y_t$ and suffers loss $\mathbb{I}[y_t \neq \hat{y}_t]$.

The notions of mistakes and regret are defined similarly and are formalized in Appendix B. The standard transductive online learning model, in which the learner has access to the concept class in advance, was studied by [BDKM97, HMS23].

# 3 Online Learning

In this section, we consider the problem of online learning with oracle access. First, we present lower bounds on the number of mistakes in the realizable setting and a lower bound on the regret in the agnostic setting with access to an ERM oracle. For the realizable case, a lower bound of $\Omega(3^{d_{\mathrm{LD}}})$ was proved by [KS24], but only for an ERM restricted to querying instances and labels generated by the adversary throughout the interaction. Here, we make no assumptions about the ERM (our result holds for the strongest "agnostic" ERM variant, see Definition 2.1), which introduces several challenges, and we show that the exponential dependence on $d_{\mathrm{LD}}$ is unavoidable. For the agnostic case, this is the first lower bound of its kind. It matches the upper bound of [AAD+23] up to a constant factor in the exponential dependence on the Littlestone dimension, and up to a $\log(T)$ factor.

**Theorem 3.1 (Lower Bound for Online Learning with "Agnostic" ERM Oracle)** *Let $\mathcal{F}$ be the family of classes with Littlestone dimension $d_{\mathrm{LD}}$. Then, any randomized algorithm that makes a finite number of queries to the ERM oracle incurs $\Omega(2^{d_{\mathrm{LD}}})$ expected mistakes.*

**Proof sketch** We construct a threshold function over $T = 2^{d_{\mathrm{LD}}}$ points embedded in $[0, 1]^{T-1}$, partitioning the space into $T$ nested hyperrectangles using uniform random values $z_1, ..., z_{T-1}$. The first point $x_1$ corresponds to the entire hypercube $[0, 1]^{T-1}$ except for a specific hyperplane, while subsequent points $x_i$ correspond to increasingly smaller nested hyperrectangles, each defined by matching more coordinates with the random values $z_j$.

The concept class consists of random threshold functions where all concepts agree within each hyperrectangle equivalence class. When the adversary presents point $x_t$, the learner cannot query points from future equivalence classes with non-zero probability, as these classes are defined by randomly chosen values $z_t, ..., z_{T-1}$ that won't be hit by finite queries. Thus, the "agnostic" ERM oracle provides labels for the current hyperrectangle but gives no information about nested hyperrectangles corresponding to future points. This geometric structure forces the learner to make predictions about points in nested regions without prior information, resulting in a mistake with probability $1/2$ at each step. This yields $\Omega(T/2) = \Omega(2^{d_{\mathrm{LD}}})$ expected mistakes. ∎

---

[4] A common version of the transductive online learning setting considers $x$ as a set of $T$ points rather than a sequence (e.g., [SKS16]). We focus on the version where $x$ is a sequence, though our results could extend to the set-based formulation.

**Theorem 3.2 (Lower Bound for Agnostic Online Learning with "Agnostic" ERM Oracle)** *Let $\mathcal{F}$ be the family of classes with Littlestone dimension $d_{\mathrm{LD}}$. Then, any randomized algorithm that makes a finite number of queries to the ERM oracle incurs $\Omega(\sqrt{T 2^{d_{\mathrm{LD}}}})$ expected regret.*

The full proofs for the lower bounds can be found in Appendix D.

We now show how any deterministic online learning algorithm using the restricted ERM can be simulated using the weak consistency oracle, at the cost of increasing the number of oracle calls by $O(T)$. In particular, this applies to the ERM-based algorithms of [AAD+23, KS24].

**Theorem 3.3 (Reducing Online Learning with Weak Consistency to Online Learning with "Restricted" ERM)** *Consider any deterministic online learning algorithm that only has access to the restricted ERM oracle. Furthermore, suppose that at each timestep $t$, for any function $f$ returned by the oracle, the algorithm evaluates $f$ only on the points $x_1, x_2, \ldots, x_t$. If this algorithm makes at most $f(T)$ mistakes and uses at most $g(T)$ oracle queries over $T$ rounds, then there exists an algorithm that uses the weak consistency oracle and makes at most $f(T)$ mistakes using at most $T \cdot g(T)$ queries.*

The proof is provided in Appendix E. As a consequence, we obtain the following result via the results in [AAD+23, KS24].

**Corollary 3.4** *There exists a learning algorithm that makes $T \cdot 2^{O(d_{\mathrm{LD}})}$ weak consistency oracle calls with $2^{O(d_{\mathrm{LD}})}$ mistakes in the realizable setting.*

Finally, we study online learning of partial concept classes. Partial concepts [AHHM22] are concepts that may be undefined on certain parts of the domain. These concepts are particularly useful for modeling data-dependent assumptions, such as the margin of the decision boundary. Recently, [DG24] showed that partial concepts can be learned in the (offline) PAC setting with the weak consistency oracle. Here, we show that such a result is impossible in the online setting, even with the ERM oracle, let alone with the weak consistency oracle. The proof is provided in Appendix F.

**Theorem 3.5 (Lower Bounds for Online Learning of Partial Concepts with ERM Oracle)** *There exists a family $\mathcal{F}$ of partial concept classes of Littlestone dimension $1$, where any algorithm that makes a finite number of queries will have $\Omega(T)$ mistakes.*

# 4 Transductive Online Learning

In this section, we study the transductive online setting with oracle access, where the main goal is to leverage the additional information given to the learner, the set of instances $x_1, \ldots, x_T$ at the start of the interaction, in order to reduce the number of mistakes or regret. The proofs for this section are provided in Appendix G.

First, we show that it is possible to recover all labelings consistent with the concept class on the given instances $x_1, \ldots, x_T$ using the weak consistency oracle. With this step, we can achieve the optimal number of mistakes in the realizable setting and optimal regret in the agnostic setting.

**Lemma 4.1 (Identify Labeling with Weak Consistency Calls)** *For a class $\mathcal{C} \subset \{0,1\}^{\mathcal{X}}$, let $d_{\mathrm{VC}}$ be the VC dimension of $\mathcal{C}$. Using the weak consistency oracle, one can recover all the concepts in $\mathcal{C}$ using $O(|\mathcal{X}|^{d_{\mathrm{VC}}+1})$ queries.*

**Theorem 4.2 (Upper Bounds for Transductive Online Learning with Weak Consistency Oracle)** *Consider any family $\mathcal{F}$ of concept classes with VC dimension $d_{\mathrm{VC}}$. There exists an algorithm that uses at most $O(T^{d_{\mathrm{VC}}+1})$ weak consistency queries and obtains optimal mistake bounds for transductive online learning (known to be upper bounded by $\min\{d_{\mathrm{LD}}, d_{\mathrm{VC}} \log T\}$), and also obtains optimal regret (known to be upper bounded by $\tilde{O}(\sqrt{T \min\{d_{\mathrm{LD}}, d_{\mathrm{VC}} \log T\}})$).*

On the other hand, we establish the following two lower bounds. The first shows that limiting the learner to $O(T)$ weak consistency queries is not sufficient for transductive online learnability, and the second shows that restricting to $O(T)$ ERM queries results in a mistake bound that is exponential in the Littlestone dimension.

**Theorem 4.3 (Lower Bound for Transductive Online Learning with Weak Consistency Oracle - Randomized Algorithms)** *Consider any family $\mathcal{F}$ of concept classes of the form $\mathcal{C} \subset \{0,1\}^{\mathcal{X}}$, where the family $\mathcal{F}$ has the property that for every labeling function $f : \{x_1, x_2, \ldots, x_T\} \to \{0,1\}$, there exists some concept class $\mathcal{C} \in \mathcal{F}$ and some concept $c \in \mathcal{C}$ such that $c(x_t) = f(x_t)$ for all $t \in [T]$ (i.e., all $2^T$ possible binary labelings of the sequence $x$ are captured by the family $\mathcal{F}$). For $T \geq 100$, any (possibly randomized) algorithm that makes at most $T/20$ queries to the weak consistency oracle will incur an expected mistake bound of at least $T/20$.*

In particular, this theorem holds for general classes like Littlestone classes, and also for special families of classes, including thresholds, $k$-intervals, and $d$-Hamming balls (see the next section).

**Proof sketch** To establish the lower bound, we consider the uniform distribution over all $2^T$ possible concept-class pairs permitted by family $\mathcal{F}$. For any deterministic algorithm making at most $T/20$ queries, we represent its execution as a binary decision tree where: (i) the root contains all $2^T$ possible concepts, (ii) internal nodes represent either oracle queries or predictions, and (iii) prediction nodes use the majority label strategy.

The expected number of mistakes plus queries equals the expected number of oracle splits and prediction mistakes on a random path from root to leaf. Transforming our tree to have a single "cost edge" to the smaller subtree at each node. We analyze the expected cost using entropy arguments. With entropy $H(X) = T$ bits, and applying the chain rule, the entropy can be expressed as the expected sum of binary entropy values $h(p(v))$ along paths from leaves to root, where $p(v)$ is the fraction of leaves below a node's smaller child, and $h(x) = -x \log_2(x) - (1-x) \log_2(1-x)$. Defining a node as "balanced" if at least $0.2$ of its leaves fall under its smaller child, entropy analysis shows there must be $\Omega(T)$ balanced nodes in expectation along a random path. Each balanced node contributes a constant fraction to the expected cost, yielding a lower bound of $\Omega(T)$, which gives us the claimed $T/20$ lower bound. ∎

**Theorem 4.4 (Lower Bound for Transductive Online Learning with ERM Oracles - Deterministic Algorithms)** *Let $\mathcal{F}$ be the family of all classes with Littlestone dimension $d_{\mathrm{LD}}$. For $T \geq 2^{d_{\mathrm{LD}}+1}$, when having access to the ERM oracle, if fewer than $T/2$ queries are made, at least $2^{d_{\mathrm{LD}}} - 1$ mistakes will be made.*

We next study specific families of concept classes, achieving stronger results than in the general case, especially using randomized algorithms.

### 4.1 Thresholds, Intervals, and Hamming Balls

We start with the family of thresholds. For a set $\mathcal{X}$, we consider the family of classes $\mathcal{F} = \{\mathcal{C}_{\preceq} \mid \preceq \text{ is a total ordering over } \mathcal{X}\}$, where each $\mathcal{C}_{\preceq}$ is the threshold class corresponding to the ordering $\preceq$. Specifically, $\mathcal{C}_{\preceq} = \{c_z : z \in \mathcal{X}\}$, where $c_z(x) = \mathbb{I}[x \preceq z]$. When restricted to $T$ points $x_1, x_2, \ldots, x_T$, the class $\mathcal{C}_{\preceq}[\{x_1, x_2, \ldots, x_T\}]$ forms a threshold class over these $T$ points, which has Littlestone dimension $\lfloor \log_2(T) \rfloor$. Our analysis focuses on both deterministic and randomized algorithms with weak consistency and ERM oracles.

**Theorem 4.5 (Upper Bounds on Transductive Online Learning of Thresholds)** *Consider transductive online learning where $\mathcal{F}$ is the family of threshold classes. The following results hold:*

1. *There exists a deterministic algorithm that makes $O(T \log T)$ calls to the weak consistency oracle and incurs at most $O(\log T)$ mistakes.*

2. *There exists a randomized algorithm that makes $O(T)$ calls in expectation to the weak consistency oracle and incurs at most $O(\log T)$ mistakes.*

3. *There exists a deterministic algorithm that makes $O(T)$ calls to the ERM oracle and incurs at most $O(\log T)$ mistakes.*

4. *There exists a randomized algorithm that makes at most $O(\log T)$ calls to the ERM oracle in expectation and makes at most $O(\log T)$ mistakes.*

We outline the proof of the last case, the full proof is in Appendix H.1.

**Proof sketch** The key observation is that when randomly sampling two points from the uncertainty region and running ERM (with one point labeled $0$ and the other labeled $1$), with constant probability, the oracle assigns label $0$ to at least $1/3$ of the points and label $1$ to at least another $1/3$. The algorithm predicts according to this labeling. When a mistake occurs, the uncertainty region shrinks by a constant factor, and we resample to generate a new labeling for the reduced region. Since the uncertainty region shrinks by a constant factor (at least $1/3$) with each mistake, we need at most $O(\log T)$ mistakes to reduce the uncertainty region to a single point. Each mistake requires only a constant number of queries in expectation (to get a "balanced" partition), leading to a total of $O(\log T)$ ERM oracle calls in expectation. ∎

We proceed to analyze another canonical family of concept classes: $k$-Intervals. For a set $\mathcal{X}$, we consider the family of classes $\mathcal{F}_{\text{int},k} = \{\mathcal{C}_{\text{int},\preceq,k} | \preceq \text{ is a total ordering over } \mathcal{X}\}$, where each $\mathcal{C}_{\text{int},\preceq,k}$ contains concepts defined by at most $k$ intervals under the ordering $\preceq$. Formally, for any total ordering $\preceq$ on $\mathcal{X}$, and any collection of at most $k$ disjoint intervals $Z_1, Z_2, \ldots, Z_m$, where $m \leq k$ and each $Z_i = \{x \in \mathcal{X} : a_i \preceq x \preceq b_i\}$ for some $a_i, b_i \in \mathcal{X}$, we define the concept as follows:

$$c_{Z_1,Z_2,\ldots,Z_m}(x) = \begin{cases} 1 & \text{if } x \in Z_i \text{ for some } i \in \{1, 2, \ldots, m\}, \\ 0 & \text{otherwise.} \end{cases}$$

This class has VC dimension $2k$ and Littlestone dimension $O(k \log T)$. It is known that this class has VC dimension $2k$, and the class defined on $T$ points has Littlestone dimension at most $T$.

**Theorem 4.6 (Upper Bound on Transductive Online Learning of $k$-Intervals with Weak Consistency Oracle - Randomized Algorithm)** *Consider transductive online learning with the family $\mathcal{F}_{int,k}$. There exists a randomized algorithm that makes $O(T^3 \cdot 2^{2k})$ calls to the weak consistency oracle in expectation and makes at most $O(k \log T)$ mistakes.*

**Proof sketch** One key observation is that for any $2k + 1$ points $z_1 \prec z_2 \prec \ldots \prec z_{2k+1} \in \mathcal{X}$, there exists exactly one non-realizable labeling, corresponding to assigning label $(i \mod 2)$ to $z_i$. This fact can be used to test with high probability whether a point $z$ is an extreme point of the ordering— repeatedly sample $2k$ other points, and if the label of $z$ is consistently $1$ for the non-realizable labeling across all samplings, then it lies on an extreme end of the threshold with high probability (after $O(T)$ samplings and $2^{2k+1}$ weak consistency oracle queries).

We apply this test to all $T$ points to identify the two endpoints of the threshold (requiring at most $O(T^2 \cdot 2^{2k+1})$ queries). We then designate one extreme as the "minimum" and recurse (at most $T$ times), yielding an upper bound of $O(T^3 \cdot 2^{2k})$ for the total number of queries. Since the Littlestone dimension is $O(k \log \frac{T}{k})$ (and the VC dimension is $O(k)$), we can run either SOA or the halving algorithm to achieve the $O(k \log T)$ mistake bound. ∎

We also study the class of $d$-Hamming Balls, which has Littlestone and VC dimensions equal to $d$. We show that an algorithm can make at most $2d$ mistakes using just a single ERM query. Furthermore, we show that the optimal mistake bound of $d$ can be achieved with $2^{d+1}$ queries. Combined with the result in Theorem 4.6, this suggests that the optimal mistake bound can be achieved for general transductive online learning using $T^C 2^{O(d_{\text{VC}})}$ queries, where $C$ is a constant independent of the class, possibly by a randomized algorithm. See Appendix H.3 for more details.

## 5 Discussion

In this paper, we studied the power and limitations of various oracles in online learning settings. In the transductive setting, there remains a gap between our general upper bound using deterministic algorithms (Theorem 4.2) and the improved performance of our randomized algorithms for special cases (Theorems 4.5 and 4.6). This motivates the following open problems:

- Given a family of classes $\mathcal{F}$ with Littlestone dimension $d_{\text{LD}}$ and VC dimension $d_{\text{VC}}$, *does there exist a randomized algorithm that uses at most $T^C 2^{O(d_{\text{VC}})}$ queries and achieves $O(d_{\text{LD}})$ or $O(d_{\text{VC}} \log T)$ expected mistakes, for some constant $C$ independent of the class?*

- *Are randomized algorithms provably more powerful than deterministic ones for online learning with oracles?*

An additional interesting direction is to extend our oracle-based framework to more general tasks, such as online multiclass classification and regression. In particular, it would be natural to design oracle-efficient algorithms for these settings, analogous to our results for the binary case. Another open question is whether the current oracle complexity bound of $T^{2^{O(d_{\mathrm{LD}})}}$ for agnostic online learning with an ERM oracle can be improved, potentially yielding more efficient algorithms in the worst-case scenario.

Our analysis thus far assumed a worst-case adversary. A relevant extension is to consider *smoothed adversaries* – adversaries with restricted power achieved by slight randomization in their choices. For example, Haghtalab et al. (2022) [HHSY22] developed the first oracle-efficient algorithm for online learning under a smoothed-adversary model, which constrains the adversary to draw each instance from a distribution of bounded density (thereby interpolating between stochastic and adversarial regimes). Block et al. (2024) [BRS24] further showed that, in such a smoothed setting, one can attain regret on the order of $\tilde{O}(\sqrt{\mathrm{comp}(\mathcal{F}) \cdot T})$ (with matching upper and lower bounds) using an ERM oracle. Here $\mathrm{comp}(\mathcal{F})$ denotes the standard PAC-learning complexity of the class $\mathcal{F}$ (e.g. VC dimension). Notably, Block et al.'s result is limited to the squared loss; extending their analysis to other loss functions (e.g. general convex losses) remains an interesting direction for future work. Finally, another avenue worth exploring is the use of weaker oracle models in conjunction with smoother adversaries. For instance, one could investigate whether a weaker consistency-based oracle (as opposed to a full ERM oracle) is sufficient to achieve low regret under smoothed adversarial conditions, potentially reducing the oracle complexity while still leveraging the adversary's restricted power.

## Acknowledgments

Idan Attias is supported by the National Science Foundation under Grant ECCS-2217023, through the Institute for Data, Econometrics, Algorithms, and Learning (IDEAL).

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

# Appendices

# A   Additional Related Work

**Oracle-Efficient Online Learning.**   Oracle-efficient methods establish a powerful framework for addressing the computational challenges of online learning. This provides an alternative to the fact that standard online learning can be computationally intractable [FL98, HBD23]. Assos et al. [AAD+23] and Kozachinskiy et al. [KS24] studied the online learnability of concept classes with finite Littlestone dimension using the ERM oracle. In addition, the partial-information setting, specifically contextual bandits, has been explored using regression oracles [FR20, SLX22] and the ERM oracle [SKS16].

Theoretical limitations of this approach have also been investigated. Hazan and Koren [HK16] showed that, for certain finite concept classes $\mathcal{C}$, any oracle-efficient online learner must make at least $\tilde{\Omega}(\sqrt{|\mathcal{C}|})$ calls to a powerful optimization oracle (an agnostic ERM oracle) to achieve sublinear regret in the fully adversarial proper-learning setting; they also give a matching $\widetilde{O}(\sqrt{|\mathcal{C}|})$ upper bound. Since the Littlestone dimension satisfies $\mathrm{Ldim}(\mathcal{C}) \leq \log_2 |\mathcal{C}|$ for finite classes, this implies an exponential dependence of total runtime on the Littlestone dimension. Our results sharpen and generalize this picture: we give lower bounds in *both* the realizable and agnostic regimes, and for more general classes parameterized by VC / Littlestone dimension. Rather than targeting mere sublinear regret, we identify conditions under which stronger, dimension-dependent rates are unattainable for any oracle-efficient learner—e.g., $\Omega(2^{d_{\mathrm{LD}}})$ (realizable) and $\Omega(\sqrt{T\,2^{d_{\mathrm{LD}}}})$ (agnostic). When specialized to finite classes with $|\mathcal{C}| \approx 2^{d_{\mathrm{LD}}}$, both lines of work exhibit exponential dependence on $d_{\mathrm{LD}}$ (since $\sqrt{|\mathcal{C}|} = 2^{d_{\mathrm{LD}}/2}$), while our results additionally showed that even in the realizable case the required computation is exponential in $d_{\mathrm{LD}}$ (and in fact cannot be done in a finite number of queries). Altogether, these findings underscore that optimization oracles remove optimization difficulty but not the information-theoretic hardness of fully adversarial online learning.

Additionally, Kalai and Vempala [KV05] and Dudík et al. [DHL+20] developed oracle-efficient algorithms for various online learning problems (e.g., combinatorial decisions and auctions) by leveraging structure or randomness (perturbed-leader methods), but their techniques assume an efficient oracle for the specific problem at hand and do not apply to arbitrary concept classes.

Several works have identified structural conditions that enable oracle-efficient online learning despite the general lower bounds. Dudík et al.[DHL+20] studied the conditions under which oracle-efficient algorithms can succeed, and Haghtalab et al. [HHSY22] provided such algorithms for online learning with smoothed adversaries, introduced in Haghtalab et al. [HRS24]. Haghtalab et al. (2022) gave the first oracle-efficient online learner in the smoothed adversary model, achieving regret bounds $O(\sqrt{T \cdot d/\sigma})$ that depend only on the hypothesis class's VC dimension $d$ and the smoothness parameter $\sigma$. Their result shows that when the adversary is constrained to draw instances from a $\sigma$-smooth distribution (i.e., no individual example can have too large a probability mass), online learning becomes computationally as easy as offline learning for any VC class. More recently, Block et al. [BRS24] considered the realizable case under smooth adversaries and show that even a simple repeated-ERM strategy can attain no-regret. In particular, they proved that if the marginal distribution is $\sigma$-smooth, then empirical risk minimization achieves sublinear error on the order of $\tilde{O}(\sqrt{\mathrm{comp}(\mathcal{F}) \cdot T})$, where $\mathrm{comp}(\mathcal{F})$ is the standard PAC-learning complexity of the class. These works underscore that smoothness assumptions, now widely used as a testbed for the robustness of impossibility results – can circumvent the brittle worst-case constructions, enabling efficient online learning of VC classes in practice. However, studying the models for general VC / Littlestone classes is necessary for gaining more fundamental insights into the computational-statistical tradeoffs that govern all concept classes. Studying general classes reveals which properties are universal versus which depend on special structure. While [HHSY22] and [HRS24] establish oracle-efficient algorithms under smoothness assumptions, oracle efficiency in the fully adversarial setting has yet to be resolved.

**PAC Learning with Weak Oracles.**   ERM has long been a foundational principle in PAC learning, particularly for binary classification, where the sample complexity is characterized by the VC dimension. However, in settings like learning partial concepts, while PAC learning is possible, it can be shown that uniform convergence, which analysis of ERM depends on, does not hold [AHHM22, Lon01]. Recently, Daskalakis et al. [DG24] introduced a weaker form of oracle, the weak consistency oracle, that suffices for learning any binary concept class with a sample complexity

that scales polynomially (roughly cubically) in $d_{\mathrm{VC}}$, where $d_{\mathrm{VC}}$ is the VC dimension. They also extend this framework to multiclass classification, regression, and partial concept classes. Their work suggests that such weak oracles could be a powerful abstraction in PAC learning, and raises the question of whether a similar approach can succeed in online learning, a question we explore in this paper.

**Online and Transductive Online Learning.** Online learning, as introduced by Littlestone [Lit88] and studied in related query models by Angluin [Ang88], is characterized by worst-case mistake bounds determined by the Littlestone dimension. The Standard Optimal Algorithm (SOA), also due to Littlestone [Lit88], achieves minimax-optimal guarantees in this model and, as such, plays a central role in the theory of online binary classification.

A related but distinct model is transductive online learning, where all unlabeled instances are revealed in advance, but labels are queried sequentially. This setting was implicitly studied in earlier work on "offline learning" [BDKM97]. The transductive online model [HMS23, HRSS24] lies between batch PAC learning and traditional online learning, and often admits more favorable learning guarantees than the latter due to the known instance sequence. Hanneke et al. [HMS23, HRSS24] provided a trichotomy characterization and multiclass extensions for this setting. Additionally, Chase et al. [CHMS25] provided a lower bound of $\Omega(\sqrt{d_{\mathrm{LD}}})$ mistakes for transductive online learning, which holds for all concept classes, and is tight (in the sense that for every integer $d$, there exists a class with Littlestone dimension $d$ for which some algorithm can achieve $O(\sqrt{d})$ mistakes).

**Batch Transductive Learning.** Batch Transductive Learning is the foundational setting introduced by Vapnik [VC⁺74, Vap82], where both labeled training data and unlabeled test instances are available at training time, and the goal is to predict labels only for the given test set rather than learn a general hypothesis. Kakade and Kalai [KK05] established connections between batch and transductive online learning, showing how efficient batch learning algorithms can be converted to efficient transductive online algorithms with regret scaling as $O(T^{3/4})$ over $T$ rounds. Cesa-Bianchi and Shamir [CBS13] improved upon this result, achieving the optimal $O(\sqrt{T})$ regret rate for a wide class of losses using a novel randomized rounding approach.

# B  Preliminaries

**Definition B.1 (Littlestone Dimension [Lit88])** Given a concept class $\mathcal{C}$, a $d$-depth Littlestone tree is a set of $\{x_{\mathbf{y}} : \mathbf{y} \in \mathcal{Y}^t, t \in \{0, d-1\}\} \subset \mathcal{X}$ (interpreting $\mathcal{Y}^0 = \{()\}$, where for all $y_1, y_2, \ldots, y_d \in \mathcal{Y}$, there's a $c \in \mathcal{C}$ such that $(c(x_{()}), c(x_{y_1}), c(x_{y_{1:2}}), \ldots, c(x_{y_{1:(d-1)}})) = (y_1, y_2, \ldots, y_d)$. The Littlestone dimension of $\mathcal{C}$ is the maximum $n$ such that there exists a Littlestone tree of depth $n$.

**Definition B.2 (VC Dimension [VC71])** Given a concept class $\mathcal{C}$, a set of $n$ points $x_1, \ldots, x_n \subset \mathcal{X}$ is shattered if $\{(c(x_1), \ldots, c(x_n)) : c \in \mathcal{C}\} = \{0, 1\}^n$. The VC dimension of $\mathcal{C}$ is the largest $n$ such that there exist $n$ shattered points in $\mathcal{X}$.

**Transductive Online Learning with Oracle Access.** The learning protocol is a sequential game between a learner and an adversary. Let $\mathcal{C} \subset \{0, 1\}^{\mathcal{X}}$ be a concept class, where $\mathcal{X}$ is the instance space and $\{0, 1\}$ is the label space. Let $\mathcal{F}$ be a family of concept classes (e.g., classes with finite Littlestone dimension) such that $\mathcal{C}$ is an unknown concept class from $\mathcal{F}$ chosen by the adversary. Suppose the learner only has oracle access to $\mathcal{C}$ via an oracle $\mathcal{O}$. First, the adversary selects a concept class $\mathcal{C} \in \mathcal{F}$ and a sequence of instances $x_{1:T} = (x_1, x_2, ..., x_T) \in \mathcal{X}^T$. The sequence $x$ is revealed to the learner[5] and the sequential game proceeds for $T$ rounds, as follows. For each $t \in [T]$:

1. The learner selects a distribution $\Delta_t \in \Delta(\{0, 1\})$, and predicts $\hat{y}_t \sim \Delta_t$.
2. The adversary reveals $y_t \in \{0, 1\}$ and the learner suffers a loss $\mathbb{I}[\hat{y}_t \neq y_t]$.

We define $y_{1:T} \in \{0, 1\}^T$ as the sequence of labels chosen by the adversary corresponding to $x_{1:T}$. In the *realizable setting*, the adversary is subject to the constraint that the sequence $(x_1, y_1), \ldots, (x_t, y_t)$

---

[5]There exist variants of this setting. One variant involves the adversary revealing a set to the learner opposed to a sequence – see [SKS16]. Our results hold for that setting as well.

is realizable by $\mathcal{C}$, meaning that there exists $c \in \mathcal{C}$ satisfying $c(x_i) = y_i$ for all $i \in [T]$. The performance of a learning algorithm $\mathcal{A}$ is measured by two metrics, the number of mistakes and the number of oracle queries. The mistakes are defined as follows:

$$M(\mathcal{A}, \mathcal{O}(\mathcal{C}), x_{1:T}, c) = \sum_{t=1}^{T} \mathbb{I}[\hat{y}_t \neq c(x_t)].$$

The worst case number of mistakes of a learning algorithm $\mathcal{A}$ is defined as

$$M(\mathcal{A}, T) = \sup_{\mathcal{C} \in \mathcal{F}} \sup_{c \in \mathcal{C}} \sup_{x_{1:T} \in \mathcal{X}^T} \mathbb{E}[M(\mathcal{A}, \mathcal{O}(\mathcal{C}), x_{1:T}, c)].$$

Similarly, $Q(\mathcal{A}, \mathcal{O}(\mathcal{C}), x_{1:T}, c)$ is the total number of queries, defined as

$$Q(\mathcal{A}, T) = \sup_{\mathcal{C} \in \mathcal{F}} \sup_{c \in \mathcal{C}} \sup_{x_{1:T} \in \mathcal{X}^T} \mathbb{E}[Q(\mathcal{A}, \mathcal{O}(\mathcal{C}), x_{1:T}, c)].$$

Here, the expectation is over the randomness of the algorithm.

We assume that the adversary is oblivious with respect to the choice of concept class $\mathcal{C}$, target concept $c$, and instance sequence $x$, meaning that these are fixed in advance and do not depend on the learner's predictions or queries.

In the agnostic case, the sequence $(x_{1:T}, y_{1:T})$ is no longer constrained to be realizable by $\mathcal{C}$, and the measure of performance is the regret, defined as

$$\text{Reg}(\mathcal{A}, \mathcal{O}(\mathcal{C}), (x_{1:T}, y_{1:T})) = \sum_{t=1}^{T} \mathbb{I}[\hat{y}_t \neq y_t] - \inf_{c \in \mathcal{C}} \sum_{t=1}^{T} \mathbb{I}[c(x_t) \neq y_t]$$

Define the worst case regret of the algorithm $\mathcal{A}$ as

$$\text{Reg}(\mathcal{A}, T) = \sup_{\mathcal{C} \in \mathcal{F}} \sup_{(x,y) \in (\mathcal{X} \times \{0,1\})^T} \mathbb{E}[\text{Reg}(\mathcal{A}, \mathcal{O}(\mathcal{C}), (x, y))],$$

and $Q(\mathcal{A}, T)$ in the agnostic case is defined as the worst-case total number of queries, over $\mathcal{C} \in \mathcal{F}$ and $(x, y) \in (\mathcal{X} \times \{0, 1\})^T$.

# C  Mistakes-Oracle Calls Pareto Frontier

## C.1  Pareto Frontier for Online Learning with ERM for Littlestone Classes

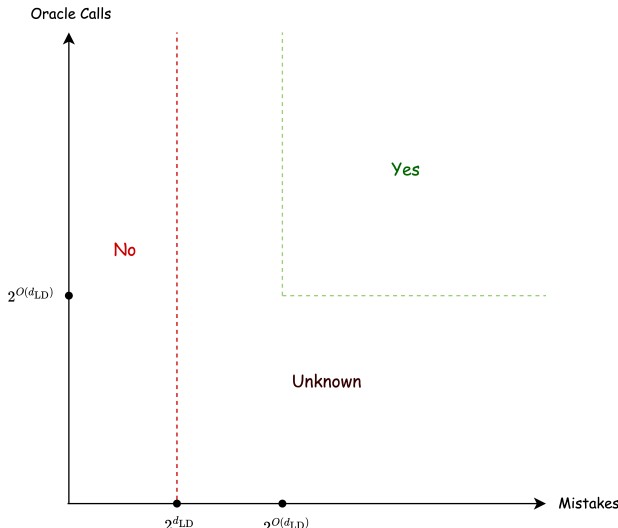

**Mistakes-Oracle Calls Pareto Frontier:
Online Learning with ERM**

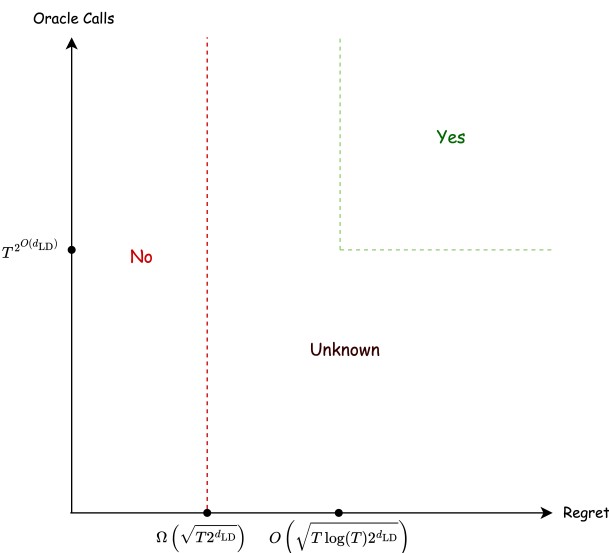

**Regret-Oracle Calls Pareto Frontier:
Agnostic Online Learning with ERM**

## C.2 Pareto Frontier for Transductive Online Learning with ERM for Littlestone Classes

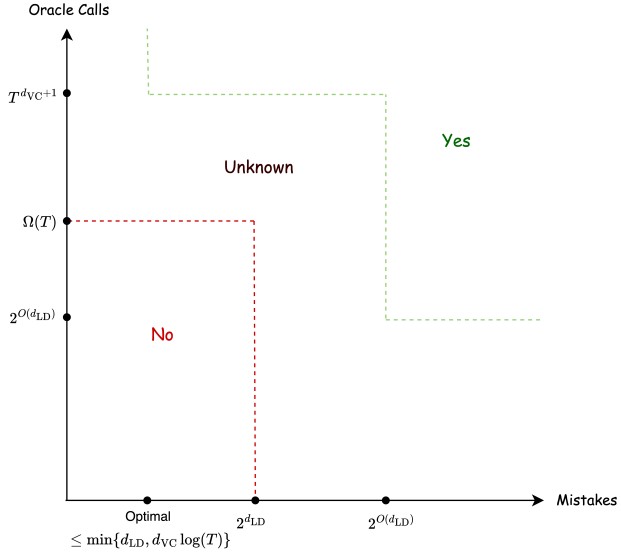

**Mistakes-Oracle Calls Pareto Frontier:**
**Transductive Online Learning with ERM**

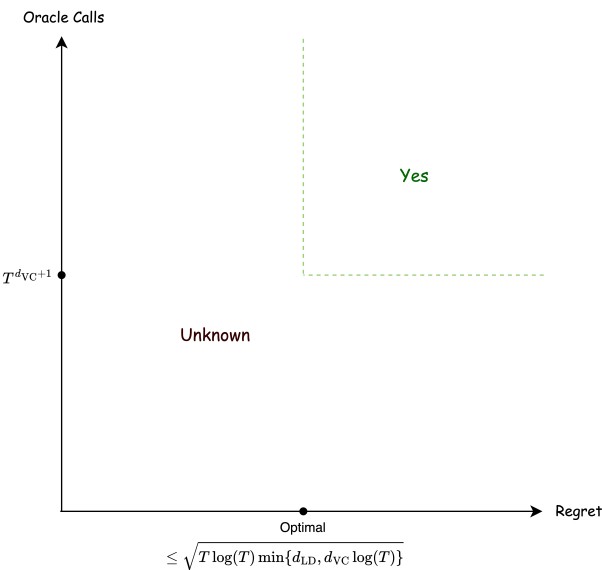

**Regret-Oracle Calls Pareto Frontier:**
**Agnostic Transductive Online Learning with ERM/Weak Consistency**

# D Online Learning with ERM Oracle: Lower Bounds for Realizable and Agnostic Settings

## D.1 ERM Oracle Lower Bound for the Realizable Case

We now cover the details for proving Theorem 3.1, the lower bound of $2^{d_{\mathrm{LD}}}$ mistakes for online learning in the realizable setting.

**Theorem 3.1 (Lower Bound for Online Learning with "Agnostic" ERM Oracle)** *Let $\mathcal{F}$ be the family of classes with Littlestone dimension $d_{\mathrm{LD}}$. Then, any randomized algorithm that makes a finite number of queries to the ERM oracle incurs $\Omega(2^{d_{\mathrm{LD}}})$ expected mistakes.*

**Proof** We prove this by constructing a hard instance where any algorithm using a finite number of ERM queries must make $\Omega(2^{d_{LD}})$ expected mistakes.

First, let $T = 2^{d_{LD}}$ and set $\mathcal{X} = [0,1]^{T-1}$. We'll define a family of concept classes parameterized by two vectors:

- $z = (z_1, \ldots, z_{T-1}) \in [0,1]^{T-1}$, which defines the geometry of our construction

- $b = (b_1, \ldots, b_T) \in \{0,1\}^T$, which defines the labeling pattern

For each $(z, b)$, we define $T$ special points in $\mathcal{X}$:

$$x_1 = (0, 0, \ldots, 0)$$
$$x_2 = (z_1, 0, \ldots, 0)$$
$$x_3 = (z_1, z_2, 0, \ldots, 0)$$
$$\vdots$$
$$x_T = (z_1, z_2, \ldots, z_{T-1})$$

These points partition $\mathcal{X}$ into $T$ "cells" $C_1, \ldots, C_T$, where:

- $C_1$ contains all points $(w_1, \ldots, w_{T-1})$ where $w_1 \neq z_1$

- $C_2$ contains all points where $w_1 = z_1$ but $w_2 \neq z_2$

- $C_i$ contains all points matching the first $i-1$ coordinates of $z$ but differing at the $i$-th coordinate

- $C_T$ contains only the point $x_T$

Now we define our concept class $\mathcal{C}_{z,b}$ as the set of all functions that:

1. Label all points in the same cell consistently

2. Assign labels to cells according to a threshold function determined by $b$

The threshold function determined by $b$ is determined as follows. $b$ determines an ordering over the points $x_1, \ldots, x_T$ in the following sense. If $b_1 = 0$, then $x_1$ is the leftmost point in the ordering, and if $b_1 = 1$, then $x_1$ is the rightmost point in the ordering. Keep repeating for $b_2, \ldots, b_T$ to construct the rest of the ordering.

The adversary works as follows:

1. Choose $z$ uniformly at random from $[0,1]^{T-1}$, and construct $x_1, x_2, \ldots, x_T$ from $z$.

2. Choose $b$ uniformly at random from $\{0,1\}^T$

3. Set the target concept $c$ such that $c(x_i) = b_i$ for all $i$.

4. Present instances $x_1, x_2, \ldots, x_T$ in order

It's sufficient to show that any algorithm incurs $\Omega(T)$ expected mistakes – by the probabilistic method, this will imply that any algorithm incurs $\Omega(T/2)$ expected mistakes on at least one instance of (concept class, target concept, first T instances).

In the first timestep, the learner is given $x_1$, which is in $C_1$. After finitely many ERM queries, the probability that any queries of the learner will be at some point $C_i$ for some $i > 1$ will be zero – this is because $z_i$ values are chosen uniformly at random from $[0,1]$, so the probability of querying any specific real value is zero. Thus, with probability 1, all the points in all the queries of the learner will be in $C_1$. The "agnostic" ERM oracle will return a concept that minimizes the error over the query points. Since the ERM oracle can return any function that minimizes this error, the adversary can choose which of these error-minimizing functions to return. Specifically, the adversary will select an error-minimizing function that provides no information about the labels of points not yet queried, especially about $x_1$. Thus, the adversary can choose the ERM to label all points in $\mathcal{X}$ as a

majority vote over the labels of the points in the query. Since $b_1$ is chosen uniformly at random from $\{0, 1\}$ and the learner has gained no information about this choice, the learner incurs a mistake with probability $1/2$.

For the $t$th timestep, the learner is given $x_t$, which is in $C_t$. After finitely many ERM queries, the probability that any queries of the learner will be at some point $C_i$ for some $i > t$ will be zero. The agnostic ERM could return any function that minimizes the error over the points in the query. Since the points in the query were only among $C_1, C_2, \ldots, C_t$, the adversary can choose the ERM to label all points in $C_t, \ldots, C_T$ the same, giving the learner no information about the locations of $C_{t+1}, \ldots, C_T$, or about $b_t$. Thus, the learner incurs a mistake with probability $1/2$ at the $t$th timestep.

Thus, over $T$ timesteps, the expected number of mistakes is $T/2$, giving the desired lower bound of $\Omega(2_{\mathrm{LD}}^d)$ expected mistakes. ∎

### D.2  ERM Oracle Lower Bound for the Agnostic Case

The lower bound for the realizable case in Theorem 3.1 can be extended to the agnostic case, with a regret that scales with $\sqrt{T}$.

**Theorem 3.2 (Lower Bound for Agnostic Online Learning with "Agnostic" ERM Oracle)** *Let $\mathcal{F}$ be the family of classes with Littlestone dimension $d_{\mathrm{LD}}$. Then, any randomized algorithm that makes a finite number of queries to the ERM oracle incurs $\Omega(\sqrt{T2^{d_{\mathrm{LD}}}})$ expected regret.*

**Proof**  We extend the construction from Theorem 3.1 to the agnostic setting.

Let $T' = 2^{d_{\mathrm{LD}}}$ and set $\mathcal{X} = [0, 1]^{T'-1}$. Choose a parameter $S$ such that $T = ST'$, where $S$ is sufficiently large for concentration bounds to apply.

We define the same special points as in the previous proof:
$$\hat{x}_1 = (0, 0, \ldots, 0)$$
$$\hat{x}_2 = (z_1, 0, \ldots, 0)$$
$$\hat{x}_3 = (z_1, z_2, 0, \ldots, 0)$$
$$\vdots$$
$$\hat{x}_{T'} = (z_1, z_2, \ldots, z_{T'-1})$$

where $z = (z_1, \ldots, z_{T'-1}) \in [0, 1]^{T'-1}$ is chosen uniformly at random.

These points partition $\mathcal{X}$ into $T'$ cells $C_1, \ldots, C_{T'}$ as defined earlier.

The adversary works as follows:

1. Choose $z$ uniformly at random from $[0, 1]^{T'-1}$, and construct $\hat{x}_1, \hat{x}_2, \ldots, \hat{x}_{T'}$.

2. For each $i = 1, \ldots, T'$ and $j = 1, \ldots, S$, present instance $x_{(i-1)S+j} = \hat{x}_i$.

3. Generate labels $y_1, y_2, \ldots, y_T$ i.i.d. uniformly from $\{0, 1\}$.

Let $b_1, b_2, \ldots b_{T'}$ be the majority of the labels in each phase (without loss of generality, break ties to favor 1s), and let the concept class be $\mathcal{C}_{z,b}$ defined in the last section.

Any algorithm will have expected error $T/2$ on this distribution (since it will make a mistake on a random label with probability $1/2$).

Now consider the performance of the best hypothesis in $\mathcal{C}_{z,b}$. This hypothesis can assign an optimal label to each cell $C_i$ based on the majority of labels seen in phase $i$. For each phase $i$ consisting of $S$ copies of $\hat{x}_i$, let $m_i$ denote the number of mistakes made by the best hypothesis. By standard concentration bounds for the binomial distribution, we have $\mathbb{E}[m_i] \leq \frac{S}{2} - c\sqrt{S}$ for some constant $c > 0$.

Summing over all $T'$ phases, the expected total error of the best hypothesis is at most
$$\frac{T}{2} - c\,T'\sqrt{S} = \frac{T}{2} - c\,2^{d_{\mathrm{LD}}}\sqrt{\frac{T}{2^{d_{\mathrm{LD}}}}} = \frac{T}{2} - c\sqrt{T2^{d_{\mathrm{LD}}}}.$$

Therefore, the expected regret of any algorithm that makes a finite number of queries to the ERM oracle over this distribution of sequences and concept classes at least $c\sqrt{T2^{d_{\mathrm{LD}}}}$, which is $\Omega(\sqrt{T2^{d_{\mathrm{LD}}}})$. By the probabilistic method, for any algorithm makes a finite number of queries to the ERM oracle, there exists a fixed concept class and sequence $(x_1, y_1), \ldots, (x_T, y_T)$ such that the expected regret is $\Omega(\sqrt{T2^{d_{\mathrm{LD}}}})$, as desired.

∎

## E  Online Learning with Weak Consistency Oracle: Upper Bounds

The following theorem enables proving upper bounds for online learning with the weak consistency oracle, assuming the existence of an ERM-based algorithm satisfying certain assumptions.

**Theorem 3.3 (Reducing Online Learning with Weak Consistency to Online Learning with "Restricted" ERM)** *Consider any deterministic online learning algorithm that only has access to the restricted ERM oracle. Furthermore, suppose that at each timestep $t$, for any function $f$ returned by the oracle, the algorithm evaluates $f$ only on the points $x_1, x_2, \ldots, x_t$. If this algorithm makes at most $f(T)$ mistakes and uses at most $g(T)$ oracle queries over $T$ rounds, then there exists an algorithm that uses the weak consistency oracle and makes at most $f(T)$ mistakes using at most $T \cdot g(T)$ queries.*

**Proof** Let $A$ be a deterministic online learning algorithm that uses a restricted ERM oracle, makes at most $f(T)$ mistakes, and uses at most $g(T)$ oracle queries over $T$ rounds. We will construct an algorithm $A'$ that uses the weak consistency oracle and maintains the same mistake bound.

At each timestep $t$, algorithm $A$ may make a query to the ERM oracle using the set $\{(x_1, y_1), (x_2, y_2), \ldots, (x_{t-1}, y_{t-1})\}$. Let's denote by $f_t$ the function that would be returned by this ERM query. By assumption, $A$ only evaluates $f_t$ on points $x_1, x_2, \ldots, x_t$.

Algorithm $A'$ will simulate $A$ as follows:

- Let $H_t$ be the set of functions returned by the ERM oracle in algorithm $A$ up to timestep $t$.

- For each function $f \in H_{t-1}$ that $A$ would evaluate at point $x_t$, algorithm $A'$ needs to determine $f(x_t)$. To do this, $A'$ will make one query to the weak consistency oracle: $\{(x_1, y_1), (x_2, y_2), \ldots, (x_{t-1}, y_{t-1}), (x_t, 0)\}$. If it returns "realizable", extend the function to have value 0 at $x_t$ and add it to $H_t$. If it returns "not realizable", extend the function to have value 1 at $x_t$ and add it to $H_t$.

- For any new ERM query that $A$ would make at timestep $t$, $A'$ needs to simulate this by finding a function consistent with $\{(x_1, y_1), (x_2, y_2), \ldots, (x_{t-1}, y_{t-1})\}$. This can be done by using the weak consistency oracle to check all possible labelings of future points that $A$ might evaluate.

The key insight is that algorithm $A$ is deterministic and, by assumption, only evaluates functions on points $x_1, x_2, \ldots, x_t$ at timestep $t$. This means that the behavior of $A$ depends only on the values of these functions on these specific points, not on their values for unseen points $x_{t+1}, \ldots, x_T$. Therefore, $A'$ makes at most $f(T)$ mistakes, just as $A$ does. Furthermore, for each ERM query in $A$, $A'$ makes at most $T$ weak consistency queries, so $A'$ makes at most $T \cdot g(T)$ weak consistency queries. ∎

We can apply the above Theorem with the algorithm by [AAD+23] to get the following.

**Corollary 3.4** *There exists a learning algorithm that makes $T \cdot 2^{O(d_{\mathrm{LD}})}$ weak consistency oracle calls with $2^{O(d_{\mathrm{LD}})}$ mistakes in the realizable setting.*

## F  Online Learning with ERM Oracle: Lower Bounds for Partial Concepts

The idea of learning partial concept classes was formalized recently by Alon, Hanneke, Holzman, and Moran [AHHM22], who demonstrated fundamental differences from the learning of total concept

classes, for example, that ERM fails to learn partial concepts, even though the VC dimension still characterizes learnability. Learning with partial concepts provides a natural framework for modeling data-dependent assumptions, such as margin conditions or cases where the target function is only defined on a subset of the input space.

Some of these ideas were previously studied implicitly in the context of regression [Lon01, BL98]. Partial concept classes have many applications, including adversarially robust learning [AKM22, AHM22], learning with fairness constraints [HP22], multiclass classification [KVK22], and online learning [CHHH23].

Let a partial concept class $\mathcal{C} \subseteq \{0, 1, \star\}^{\mathcal{X}}$. For $c \in \mathcal{C}$ and input $x$ such that $c(x) = \star$, we say that $c$ is *undefined* on $x$. The *support* of a partial concept $c : \mathcal{X} \to \{0, 1, \star\}$ is $\text{supp}(c) = \{x \in \mathcal{X} : c(x) \neq \star\}$. A sequence $S = ((x_1, y_1), \ldots, (x_m, y_m))$ is *realizable* by $\mathcal{C}$ if there exists $c \in \mathcal{C}$ such that $c(x_i) = y_i$ for all $i \in [m]$ and $x_i \in \text{supp}(c)$ for all $i \in [m]$.

The *Littlestone dimension* of a partial class $\mathcal{C}$ is the maximum depth $d$ such that there exists a $d$-depth Littlestone tree where for each path from root to leaf, there exists some concept $c \in \mathcal{C}$ that realizes the path and is defined on all points along that path (see Definition B.1 for the definition of a Littlestone tree). The *VC dimension* of a partial class $\mathcal{C}$ is defined as the maximum size of a shattered set $S \subseteq \mathcal{X}$, where $S$ is shattered by $\mathcal{C}$ if the projection of $\mathcal{C}$ on $S$ contains all possible binary patterns: $\{0, 1\}^S \subseteq \mathcal{C}|_S$.

**Theorem 3.5 (Lower Bounds for Online Learning of Partial Concepts with ERM Oracle)** *There exists a family $\mathcal{F}$ of partial concept classes of Littlestone dimension $1$, where any algorithm that makes a finite number of queries will have $\Omega(T)$ mistakes.*

**Proof** Let $\mathcal{X} = [0, 1]$, Construct a family of partial concept classes $\mathcal{F}$ as follows:

Letting $z = (z_1, z_2, \ldots, z_T)$, and $b \in \{0, 1\}^T$, define $\mathcal{C}_{z,b}$ as a collection of the functions $c_1, c_2, \ldots c_T$ where

$$\begin{cases} c_i(z_j) = b_j & \text{if } j < i \\ c_i(z_j) = 1 - b_j & \text{if } j = i \\ c_i(x) = \star & \text{if } x \notin \{z_1, z_2, \ldots, z_i\} . \end{cases}$$

The Littlestone dimension of $\mathcal{C}_{z,b}$ is $1$.

The strategy of the adversary is as follows:

- Choose $z$ and $b$ (sample $z_i$ uniformly from $[0, 1]$ and $b_i$ uniformly from $\{0, 1\}$ for all $i$).

- At each timestep $t$, present point $x_t = z_t$ to the algorithm

At timestep $t$, when the learner makes a query to the ERM oracle, the learner won't discover any of the points among $x_{t+1}, x_{t+2}, \ldots, x_T$ with probability $1$, and thus, the queries will only be among points in $\mathcal{X} - \{x_{t+1}, x_{t+2}, \ldots, x_T\}$. If any points outside of $\{x_1, \ldots, x_T\}$ the ERM oracle will return "not realizable" (since any target concept will return $\star$ on such points).

Thus, suppose the queries are all among points in $\{x_1, \ldots, x_T\}$. When the learner makes ERM queries on any subset of points from $\{z_1, z_2, \ldots, z_t\}$, there are always two valid extensions of the concept - one where $b_t = 0$ and one where $b_t = 1$. This is because:

1. For any $j < t$, both $c_t$ and $c_{t+1}$ agree on $z_j$ (both output $b_j$).

2. For $z_t$ itself, $c_t$ outputs $1 - b_t$ while $c_{t+1}$ outputs $b_t$.

Since this is the case, the learner won't gain any information about $b_t$, so the adversary will make a mistake with probability $1/2$ at each timestep since $b_t$ is chosen uniformly at random from $\{0, 1\}$, independently of the learner's prediction. Thus, over $T$ timesteps, the expected number of mistakes is $T/2$, giving the desired lower bound of $\Omega(T)$ expected mistakes. ∎

# G Transductive Online Learning: General Concept Classes

Here we present the details for upper bounds and lower bounds for transductive online learning on general concept classes.

## G.1 Upper Bounds with the Weak Consistency Oracle

We first start with a lemma which discusses a "preprocessing" step that the learner can choose to perform to gain knowledge of the concept class.

**Lemma 4.1 (Identify Labeling with Weak Consistency Calls)** *For a class $\mathcal{C} \subset \{0,1\}^{\mathcal{X}}$, let $d_{\mathrm{VC}}$ be the VC dimension of $\mathcal{C}$. Using the weak consistency oracle, one can recover all the concepts in $\mathcal{C}$ using $O(|\mathcal{X}|^{d_{\mathrm{VC}}+1})$ queries.*

**Proof** Let $N = |\mathcal{X}|$. Let $\mathcal{X}$ be expressed as $x_1, x_2, \ldots x_N$. Since $\mathcal{C}$ has VC dimension $d_{\mathrm{VC}}$, the number of distinct labelings it can realize on $N$ points is $O(N^{d_{\mathrm{VC}}})$ by Sauer's Lemma [SSBD14].

For each point $x_i$, where $i \in [N]$, we check which labelings are consistent with the data seen so far. Specifically, for each prefix $(x_1, y_1), \ldots, (x_{i-1}, y_{i-1})$ that has been determined to be realizable, we query the weak consistency oracle twice: once with the additional labeled example $(x_i, 0)$ and once with $(x_i, 1)$. We maintain a tree of realizable labeled prefixes, expanding it level by level.

At each level, each realizable prefix may branch into two new prefixes, depending on whether labeling $x_i$ with 0 or 1 is consistent. Since the total number of distinct full labelings is $O(N^{d_{\mathrm{VC}}})$, there are at most $O(N \cdot N^{d_{\mathrm{VC}}})$ possible prefixes. Since each step involves up to two queries per prefix, the total number of oracle calls is at most $O(N^{d_{\mathrm{VC}}+1})$. ∎

**Theorem 4.2 (Upper Bounds for Transductive Online Learning with Weak Consistency Oracle)** *Consider any family $\mathcal{F}$ of concept classes with VC dimension $d_{\mathrm{VC}}$. There exists an algorithm that uses at most $O(T^{d_{\mathrm{VC}}+1})$ weak consistency queries and obtains optimal mistake bounds for transductive online learning (known to be upper bounded by $\min\{d_{\mathrm{LD}}, d_{\mathrm{VC}} \log T\}$), and also obtains optimal regret (known to be upper bounded by $\tilde{O}(\sqrt{T \min\{d_{\mathrm{LD}}, d_{\mathrm{VC}} \log T\}})$).*

**Proof** Let $\mathcal{C}$ be the unknown concept class. Consider $\mathcal{C}[\{x_1, \ldots, x_T\}]$, the class of $\mathcal{C}$ restricted to $\{x_1, \ldots, x_T\}$. This class has VC dimension at most $d_{\mathrm{VC}}$. Apply this to Lemma 4.1 to get all realizable labelings using $O(T^{d_{\mathrm{VC}}+1})$ weak consistency queries.

Once we have identified all realizable labelings on $\{x_1, \ldots, x_T\}$, we are now in the transductive online learning setting where the class is known. Thus, one can use any standard (non-oracle) transductive online learning algorithm to get a mistake bound equivalent to in the non-oracle setting, for example by using the SOA algorithm [Lit88] (to get an $O(d_{\mathrm{LD}})$ mistake bound) or a halving algorithm on the number of shattered sets [KK05] (to get an $O(d_{\mathrm{VC}} \log T)$ mistake bound). ∎

## G.2 A Lower Bound with the Weak Consistency Oracle

We prove a general randomized lower bound of $\Omega(T)$ queries needed to attain $o(T)$ regret for the setting with the weak consistency oracle. We express it via the following lemma.

**Theorem 4.3 (Lower Bound for Transductive Online Learning with Weak Consistency Oracle - Randomized Algorithms)** *Consider any family $\mathcal{F}$ of concept classes of the form $\mathcal{C} \subset \{0,1\}^{\mathcal{X}}$, where the family $\mathcal{F}$ has the property that for every labeling function $f : \{x_1, x_2, \ldots, x_T\} \to \{0,1\}$, there exists some concept class $\mathcal{C} \in \mathcal{F}$ and some concept $c \in \mathcal{C}$ such that $c(x_t) = f(x_t)$ for all $t \in [T]$ (i.e., all $2^T$ possible binary labelings of the sequence $x$ are captured by the family $\mathcal{F}$). For $T \geq 100$, any (possibly randomized) algorithm that makes at most $T/20$ queries to the weak consistency oracle will incur an expected mistake bound of at least $T/20$.*

Informally, with the weak consistency oracle, $O(T)$ queries correspond to $\Omega(T)$ mistakes for any concept class for any family of classes that capture all $2^T$ labelings. Notice here that this allows us to have a lower bound for general families of classes, including general Littlestone classes and

general VC classes. Additionally, by the conditions in the theorem, this lower bound holds for settings including those in Appendix H.1(thresholds, $k$-intervals) and Appendix H.3 ($d$-Hamming balls).

To prove Theorem 4.3, we make use of the following lower bound.

**Lemma G.1 (Lower Bound for Binary Tree Costs)** *There exists a universal constant $C$ such that the following is true. Given an integer $n > 10$, construct any binary tree, with labeled nodes such that the following conditions are satisfied:*

- *The root node has value $n$*

- *For any node with value $x$ strictly greater than $1$, its two children have positive integer values $y$ and $z$ such that $x = y + z$.*

- *The depth of the tree is at most $1.05 \log_2(n)$*

*For an internal node with children $y$ and $z$, define the cost of the node to be $\min(y, z)$. Define the cost of the tree as the sum of the costs of the internal nodes. Then, the tree has cost $\geq C \log_2(n)$.*

**Proof** Consider the random variable $X$, which is taken uniformly at random over all the leaves. The idea is to relate the entropy $H(X)$ to the cost.

For each internal node $v$, define $p(v)$ to be the fraction of leaves in its subtree that are in the subtree of the smaller child:

$$p(v) = \min(|v_L|, |v_R|)/|v|$$

where $v_L$ and $v_R$ are the left and right children of $v$. Additionally, let $h(x)$ be the binary-entropy function ($h(x) = -x \log_2 x - (1 - x) \log_2(1 - x)$).

By the chain rule of entropy, we can decompose $H(X)$ as follows. For the root node $r$, let $X_r$ be the random variable corresponding to which side of the root $X$ will lie on ($X_r = 0$ if it lies on the subtree of the smaller child, and $X_r = 1$ otherwise). The chain rule ($H(Z_1, Z_2) = H(Z_1) + H(Z_2|Z_1)$) gives that:

$$H(X) = H(X_r) + H(X|X_r) = h(p(r)) + \mathbb{E}_{c \sim X_r}[H(X|X_r = c)]$$

$$= h(p(r)) + p(r)H(X|X_r = 0) + (1 - p(r))H(X|X_r = 1)$$

We can continue this decomposition recursively through the tree. At each internal node $v$, we get a term $h(p(v))$ weighted by the probability that $X$ reaches node $v$, which is $\frac{|v|}{n}$. Thus, we have:

$$H(X) = \sum_{v \text{ internal}} \frac{|v|}{n} h(p(v))$$

Equivalently, we can view this summation from the perspective of a randomly chosen leaf. For each leaf $l$, let $\text{path}(l)$ be the set of internal nodes on the path from the root to $l$. Then:

$$H(X) = \mathbb{E}_{l \sim X}\left[ \sum_{v \in \text{path}(l)} h(p(v)) \right]$$

.

Since $X$ is uniformly distributed over the leaves, each leaf contributes equally to this expectation.

Also the cost of the tree (divided by $n$) is equal to

$$\sum_{v} \frac{|v|}{n} p(v) = \mathbb{E}_{l \sim X}\left[ \sum_{v \in \text{path}(l)} p(v) \right]$$

.

Consider a threshold $t$, and define a node $v$ to be unbalanced if $p(v) < t$, and balanced otherwise. For balanced nodes, we have $h(p(v)) \leq 1$, and for unbalanced nodes, we have $h(p(v)) \leq h(t)$. Let

$b(l)$ and $u(l)$ be the number of balanced and unbalanced nodes, respectively, on the path from the root to leaf $l$.

Since $b(l) + u(l) \leq 1.05 \log_2(n)$ by the depth constraint, the above is at most:

$$\mathbb{E}_{l \sim X}[b(l) + (1.05 \log_2(n) - b(l))h(t)]$$

Setting $\log_2 n \leq$ this and rearranging gives:

$$\log_2 n(1 - 1.05h(t)) \leq \mathbb{E}_{l \sim X}[b(l)(1 - h(t))]$$

$$\Rightarrow \mathbb{E}_{l \sim X}[b(l)] \geq \log_2(n)\frac{1 - 1.05h(t)}{1 - h(t)}$$

Additionally, the cost divided by $n$ is at least:

$$\frac{\text{cost}}{n} \geq \mathbb{E}_{l \sim X}[t \cdot b(l)]$$

This is because each balanced node $v$ contributes at least $t$ to the cost ratio (by definition of being balanced).

Combining these results gives that the cost is at least:

$$\text{cost} \geq n \log_2(n)\frac{t(1 - 1.05h(t))}{1 - h(t)}$$

Setting $t = 0.2$ gives that the cost is at least $0.1n \log_2 n$, which establishes our claim with constant $C = 0.1$.

∎

**Proof of Theorem 4.3** Consider all $2^T$ concepts over $\{x_1, \ldots, x_T\}$, each paired with a concept class that contains it, permitted by the family $\mathcal{F}$. Consider the uniform distribution over all these $2^T$ (concept, concept class) pairs. Consider any deterministic algorithm that makes at most $T/20$ queries on any input. For any deterministic algorithm that makes at most $T/20$ queries on any input, we will show that the expected number of mistakes plus queries is at least $T/10$.

We can represent the algorithm's execution as a binary decision tree. The root node contains all $2^T$ possible (concept, concept class) pairs. The tree branches in one of two ways:

1. At a query node: The algorithm queries the weak consistency oracle, creating two children corresponding to the possible responses (yes/no). Both edges are colored red to represent oracle queries.

2. At a prediction node: The algorithm makes a prediction for some example. To minimize mistakes, the optimal strategy is to predict the majority label among remaining concepts. When a mistake occurs (prediction differs from the true concept's label), we draw a blue edge to the corresponding child node. This child will contain fewer leaves than its sibling, as it represents the minority prediction.

The leaf nodes each correspond to a single (concept, concept class) pair, uniquely identified through the algorithm's execution path.

The expected cost (mistakes plus queries) equals the expected number of red and blue edges encountered on a random path from root to leaf, where the randomness comes from uniformly selecting among all possible concepts. This cost can be lower-bounded by considering a modified tree where each query node has only one red edge pointing to the child with fewer leaves (rather than two red edges to both children).

By applying Lemma G.1 to this modified tree, with each node labeled by the number of leaves in its subtree, we obtain a lower bound of $\Omega(T)$ on the expected cost. Specifically, with the constant derived in Lemma G.1, the expected number of mistakes plus queries is at least $T/10$.

Since a randomized algorithm is simply a distribution over deterministic algorithms, by Yao's minimax principle, the expected cost of any randomized algorithm against our uniform distribution

over concepts is also at least $T/10$. Therefore, any algorithm making at most $T/20$ queries must incur at least $T/20$ expected mistakes.

∎

### G.3 A Lower Bound with the ERM Oracle

A lower bound for mistakes that is exponential in the Littlestone dimension can additionally be shown for the ERM Oracle in the transductive online setting. The following theorem complements Theorem 3.1 from the online setting, but is a lower bound on deterministic algorithms, with an assumption on the number of queries made ($o(T)$).

**Theorem 4.4 (Lower Bound for Transductive Online Learning with ERM Oracles - Deterministic Algorithms)** *Let $\mathcal{F}$ be the family of all classes with Littlestone dimension $d_{\mathrm{LD}}$. For $T \geq 2^{d_{\mathrm{LD}}+1}$, when having access to the ERM oracle, if fewer than $T/2$ queries are made, at least $2^{d_{\mathrm{LD}}} - 1$ mistakes will be made.*

**Proof** We consider the family of classes $\mathcal{F}_{\mathrm{tr},k}$, which consists of threshold functions partitioned into $k$ equivalence classes, defined as follows:

For every partition of $\mathcal{X}$ into $\mathcal{X}_1 \sqcup \mathcal{X}_2 \sqcup \ldots \sqcup \mathcal{X}_k$, we construct a class $\mathcal{C}$ containing $k + 1$ concepts $c_0, c_1, \ldots, c_k$, where:

$$c_i(x) = \begin{cases} 0 & \text{if } x \in \mathcal{X}_j \text{ for some } j \leq i \\ 1 & \text{otherwise} \end{cases}$$

We specifically consider $\mathcal{F}_{\mathrm{tr},2^{d_{\mathrm{LD}}}}$. All concepts in this family have Littlestone dimension at most $d_{\mathrm{LD}}$, so $\mathcal{F}_{\mathrm{tr},2^{d_{\mathrm{LD}}}} \subset \mathcal{F}$. Therefore, proving a lower bound for $\mathcal{F}_{\mathrm{tr},2^{d_{\mathrm{LD}}}}$ will establish a lower bound for $\mathcal{F}$.

The key property is that classes in this family can be viewed as thresholds on $2^{d_{\mathrm{LD}}}$ points, where the points are divided into $2^{d_{\mathrm{LD}}}$ equivalence classes. Our adversarial strategy will reveal information about these equivalence classes one at a time, with each mistake forcing the learner to move to a new equivalence class.

For each timestep $t = 1, \ldots, T$, after $q$ queries have been processed during timestep $t$, we maintain three sets:

- $O_t^{(q)}$: Old threshold points whose labels are definitively known

- $C_t^{(q)}$: Copies of the current threshold point (all having the same known label)

- $U_t^{(q)}$: Uncertainty region (points whose labels are unknown)

We initialize $O_1^{(0)} := \emptyset, C_1^{(0)} := \emptyset, U_1^{(0)} = \{x_1, x_2, \ldots, x_T\}$.

Let $K$ be an integer denoting how many equivalence classes of the threshold have been currently observed. Initially, $K = 0$.

At the beginning of timestep 1, the learning algorithm will either make a prediction or a query:

- If a query is made: The learner cannot gain information by assigning the same label to all points in the query. Thus, the query must contain both a point labeled 0 and a point labeled 1. Let $z_0$ be the earliest indexed point in the query labeled 0, and let $z_1$ be the earliest indexed point labeled 1.

  The adversary returns "not realizable" and designates $z_1$ as the first point of the first equivalence class with label one on the extreme right end of the threshold (increment $K$ to 1 and set $y_c = 1$). Update $C_1^{(1)} := \{z_1\}, O_1^{(1)} := \emptyset$, and $U_1^{(1)} = U_1^{(0)} \setminus \{z_1\}$.

- If a prediction $\hat{y}_1$ is made for $x_1$: The adversary makes this prediction incorrect by setting $y_1 = 1 - \hat{y}_1$, and designates $x_1$ as the first point of the first equivalence class, and places $x_1$

on the extreme end of the threshold corresponding to $y_1$ (increment $K$ to 1 and set $y_c = y_1$). Update $C_1^{(1)} := \{x_1\}$, $O_1^{(1)} := \emptyset$, and $U_1^{(1)} = U_1^{(0)} \setminus \{x_1\}$.

Let $K$ be an integer, which denotes how many equivalence classes of the threshold have been currently seen. After the above, only (part of) one equivalence class has been seen, so set $K = 1$.

For subsequent operations, suppose we are at timestep $t$, having made $q$ queries in this timestep. This corresponds to sets $O_t^{(q)}, C_t^{(q)}, U_t^{(q)}$, where points in $C_t^{(q)}$ have label $y_c$ and belong to the $K$-th equivalence class. The algorithm proceeds as follows:

- If the learner makes a query:

  - Including points from $O_t^{(q)}$ in queries provides no additional information. This is because points in $O_t^{(q)}$ have fixed, known labels that constrain the possible labelings of $U_t^{(q)}$, but do not provide new information beyond what is already known.

  - Suppose the query includes points from $C_t^{(q)}$ and $U_t^{(q)}$. The query points from $C_t^{(q)}$ must always have label equal to $y_c$ (otherwise, all points in $U_t^{(q)}$ will be forced to have label $1 - y_c$, in which case the queries will give no information). Furthermore, for the query to provide new information, there must exist some point $z \in U_t^{(q)}$ assigned label $1 - y_c$ in the query.
    The adversary returns "not realizable" and selects $z$ to be the earliest indexed point in $U_t^{(q)}$ labeled $1 - y_c$ in the query. The adversary then designates $z$ as another point in the current equivalence class: $C_t^{(q+1)} := C_t^{(q)} \cup \{z\}$, $U_t^{(q+1)} := U_t^{(q)} \setminus \{z\}$, $O_t^{(q+1)} := O_t^{(q)}$.

  - If the query includes only points from $U_t^{(q)}$: If all labels in the query are identical, the ERM can satisfy the query without providing useful information (by labeling all elements in $U_t^{(q)}$ the same. Therefore, the query must include both a point labeled 0 and a point labeled 1.
    Let $z_0$ be the earliest indexed element of $U_t^{(q)}$ labeled 0 in the query, and let $z_1$ be the earliest indexed element labeled 1. The adversary returns "not realizable" and designates $z_{1-y_c}$ as a copy of the elements in $C_t^{(q)}$: $C_t^{(q+1)} := C_t^{(q)} \cup \{z_{1-y_c}\}$, $U_t^{(q+1)} := U_t^{(q)} \setminus \{z_{1-y_c}\}$, $O_t^{(q+1)} := O_t^{(q)}$.

- If the learner makes a prediction for $x_t$:

  - If $x_t \in O_t^{(q)} \cup C_t^{(q)}$, then the learner already knows the correct label. Update $O_{t+1}^{(0)} := O_t^{(q)}$, $C_{t+1}^{(0)} := C_t^{(q)}$, $U_{t+1}^{(0)} := U_t^{(q)}$.

  - If $x_t \in U_t^{(q)}$ and the learner predicts $\hat{y}_t$: The adversary sets $y_t = 1 - \hat{y}_t$, forcing a mistake, and increments $K$ by 1 (moving to a new equivalence class, and placing $x_t$ on an extreme end of $U_t^{(q)}$ in the ordering corresponding to $y_t$). The point $x_t$ becomes the first representative of this new class.
    If $K < 2^{d_{\mathrm{LD}}}$, update $C_{t+1}^{(0)} := \{x_t\}$, $U_{t+1}^{(0)} := U_t^{(q)} \setminus \{x_t\}$, $O_{t+1}^{(0)} := O_t^{(q)} \cup C_t^{(q)}$, and set $y_c = y_t$ for the new equivalence class.
    If $K = 2^{d_{\mathrm{LD}}}$, then all remaining points in $U_t^{(q)}$ belong to the same final equivalence class with label $y_t$. Update $O_{t+1}^{(0)} = \{x_1, \ldots, x_T\}$, $C_{t+1}^{(0)} = \emptyset$, $U_{t+1}^{(0)} = \emptyset$.

An essential property is that each query provides information about at most one point. Since the learner makes fewer than $T/2$ queries, there are at least $\lceil T/2 \rceil$ points about which no information is gained through queries. Let these points, in order of their indices, be $z_1, z_2, \ldots, z_{\lceil T/2 \rceil}$.

Given that $T/2 \geq 2^{d_{\mathrm{LD}}}$, the adversary can force mistakes on at least the first $2^{d_{\mathrm{LD}}} - 1$ of these points, with each mistake corresponding to the discovery of a new equivalence class. Therefore, the learner makes at least $2^{d_{\mathrm{LD}}} - 1$ mistakes.

■

# H  Transductive Online Learning: Thresholds, Intervals, and Hamming Balls

To gain deeper insight into learning Littlestone classes using ERM oracle queries, we analyze two characteristic examples of Littlestone classes: $d$-Hamming Balls and thresholds. We additionally study unions of $k$ intervals.

## H.1  Thresholds

We restate our results for transductive online learning with thresholds.

**Theorem 4.5 (Upper Bounds on Transductive Online Learning of Thresholds)**  *Consider transductive online learning where $\mathcal{F}$ is the family of threshold classes. The following results hold:*

1. *There exists a deterministic algorithm that makes $O(T \log T)$ calls to the weak consistency oracle and incurs at most $O(\log T)$ mistakes.*

2. *There exists a randomized algorithm that makes $O(T)$ calls in expectation to the weak consistency oracle and incurs at most $O(\log T)$ mistakes.*

3. *There exists a deterministic algorithm that makes $O(T)$ calls to the ERM oracle and incurs at most $O(\log T)$ mistakes.*

4. *There exists a randomized algorithm that makes at most $O(\log T)$ calls to the ERM oracle in expectation and makes at most $O(\log T)$ mistakes.*

**Proof  Deterministic + weak consistency oracle (Part 1):** The key insight is that for any two points $x_i, x_j$, we can determine their relative ordering in $\preceq$ using a single query to the weak consistency oracle. Specifically, we check if the dataset $\{(x_i, 0), (x_j, 1)\}$ is realizable by $\mathcal{C}_{\preceq}$. If it is realizable, then $x_i \preceq x_j$; otherwise, $x_j \preceq x_i$.

To determine the complete ordering of all $T$ points, we employ a comparison-based sorting algorithm requiring $O(T \log T)$ oracle calls. Once the points are sorted according to $\preceq$, we can run standard online learning algorithms (such as the Standard Optimal Algorithm or the Halving Algorithm) on the sorted sequence, incurring at most $O(\log T)$ mistakes, which matches the Littlestone dimension of the class.

**Randomized + weak consistency oracle (Part 2):** We present a randomized algorithm that efficiently learns the threshold by maintaining boundary points and strategically sampling points with unknown labels.

For each time step $t$, we maintain two boundary points:

- $r_t$: the rightmost point in $\{x_1, x_2, \ldots, x_{t-1}\}$ labeled 0

- $\ell_t$: the leftmost point in $\{x_1, x_2, \ldots, x_{t-1}\}$ labeled 1

Initially, we set $r_1 = -\infty$ and $\ell_1 = \infty$ as sentinel values to represent the entire range.

When presented with a new point $x_t$, we first determine its position relative to our current boundaries using two weak consistency oracle queries:

- Query 1: Is $\{(x_t, 1), (r_t, 0)\}$ realizable? If not realizable, then $x_t \preceq r_t$, so $x_t$ must be labeled 0.

- Query 2: Is $\{(\ell_t, 1), (x_t, 0)\}$ realizable? If not realizable, then $\ell_t \preceq x_t$, so $x_t$ must be labeled 1.

If either query resolves the label, we predict accordingly. Otherwise, $x_t$ falls in the uncertainty region $U_t = (r_t, \ell_t)$. To predict efficiently, we estimate $x_t$'s relative position within $U_t$ as follows:

In the initial case when $r_1 = -\infty$ and $\ell_1 = \infty$, these comparisons are not well-defined. For the first few points until both boundary points are properly established, we employ a simple strategy: predict 0 until observing the first 1, then predict 1 until observing the first 0 (if ever). This approach guarantees at most 2 mistakes during this initialization phase, after which our boundary points become well-defined and the main algorithm takes over.

We sample $m = \lceil 36 \log(1/\delta) \rceil$ points uniformly at random from $U_t$, where $\delta$ is a small constant probability of error. For each sampled point $z_i$, we query whether $z_i \preceq x_t$ and record the indicator $B_i = \mathbf{1}[z_i \preceq x_t]$. Let $\hat{p} = \frac{1}{m} \sum_{i=1}^{m} B_i$ be our estimate of the true fraction $p = \mathbb{P}_{z \sim U_t}[z \preceq x_t]$, which represents $x_t$'s relative position in $U_t$.

By Hoeffding's inequality:

$$\mathbb{P}[|\hat{p} - p| > 1/6] \leq 2 \exp(-2m(1/6)^2) \leq \delta$$

Therefore, with probability at least $1 - \delta$, we have one of the following cases:

- If $\hat{p} \leq 1/3$: $x_t$ likely lies in the first third of $U_t$, so we predict label 0

- If $\hat{p} \geq 2/3$: $x_t$ likely lies in the last third of $U_t$, so we predict label 1

- Otherwise: We can predict either label (for concreteness, predict 0)

If our prediction is incorrect, the threshold must lie in at most $2/3$ of the original interval $U_t$. After each mistake, we update $r_t$ or $\ell_t$ based on the observed label, further constraining the uncertainty region. Since each mistake reduces the length of the uncertainty region by a factor of at least $3/2$, the total number of mistakes is bounded by $\lceil \log_{3/2} T \rceil = O(\log T)$. Once the true label $y_t$ is known, one can perform a constant number of queries to update $r_{t+1}$ and $\ell_{t+1}$ based on the value of $y_t$.

In practice, we cannot directly sample from $U_t$ since we don't know which unlabeled points lie within it. Instead, we sample from all unlabeled points and test each sampled point $z$ using the weak consistency oracle:

- Query if $z \preceq r_t$: If true, $z$ is to the left of $U_t$ (and label $z$ as 1 and discard it from the set of unlabeled points)

- Query if $\ell_t \preceq z$: If true, $z$ is to the right of $U_t$ (and label $z$ as 0 and discard it from the set of unlabeled points)

- Otherwise, $z \in U_t$ and can be used in our estimation

Samplings of the first and second types will happen at most once, since it can only be discarded at most once from the set of unlabeled points. Thus, the first and second types of samples contribute $O(T)$ calls to the weak consistency oracle. The analysis for the third type carries from earlier. Thus, the total number of oracle calls is $O(T)$, with $O(\log T)$ mistakes.

**Deterministic + ERM Oracle (Part 3):** We present a deterministic algorithm that uses the ERM oracle to efficiently learn the total ordering of all points, then applies a standard online learning algorithm.

First, we employ a divide-and-conquer approach to sort all points:

1. Select two arbitrary points $z_1$ and $z_2$ from the current set.

2. Make two ERM oracle queries: one with the labeled set $\{(z_1, 0), (z_2, 1)\}$ and another with $\{(z_1, 1), (z_2, 0)\}$.

3. Exactly one of these queries will be realizable by the threshold class. If $\{(z_1, 0), (z_2, 1)\}$ is realizable, then $z_1 \preceq z_2$; otherwise, $z_2 \preceq z_1$.

4. Consider the concept $c$ defined by the ERM oracle query that is realizable. This concept partitions the points into two sets: $S_0 = \{x : c(x) = 0\}$ and $S_1 = \{x : c(x) = 1\}$.

5. By the nature of threshold functions, all points in $S_0$ must precede all points in $S_1$ in the underlying total order. Recursively sort $S_0$ and $S_1$ using the same approach.

For the base case (when a subset has size 1 or 0), no sorting is needed.

Let $T(n)$ be the number of ERM oracle calls needed to sort $n$ points. We have: $T(n) = T(|S_0|) + T(|S_1|) + 2$, where $|S_0| + |S_1| = n$

In the worst case, this recurrence solves to $T(n) = O(n)$, meaning our algorithm makes $O(T)$ calls to the ERM oracle.

Once the points are sorted, we can run a standard online algorithm just as was done in the (deterministic + weak consistency oracle) case, incurring at most $O(\log T)$ mistakes.

**Randomized + ERM Oracle (Part 4):** We present a randomized algorithm that efficiently learns the threshold by maintaining and updating a partitioning of the input space.

We maintain three disjoint sets throughout the algorithm:

- $L_t$: Points known to be to the left of the threshold (labeled 0)

- $R_t$: Points known to be to the right of the threshold (labeled 1)

- $U_t$: Points with unknown labels (i.e., points between $L_t$ and $R_t$)

Initially, we set $L_1 = \emptyset$, $R_1 = \emptyset$, and $U_1 = \{x_1, x_2, \ldots, x_T\}$.

At the beginning of each time step $t$, if we need to make a prediction for point $x_t$, we first check if we already know its label:

- If $x_t \in L_t$, predict $\hat{y}_t = 0$

- If $x_t \in R_t$, predict $\hat{y}_t = 1$

If $x_t \in U_t$, we need to make a prediction based on our current understanding of the threshold location. We proceed as follows:

1. **Base Case:** If $|U_t| \leq 5$, we determine the exact ordering of the remaining points with a constant number of ERM oracle calls using a simple sorting algorithm, and predict accordingly.

2. **Partitioning Step:** If $|U_t| > 5$ and we haven't already computed a partition $(U_{L,t}, U_{R,t})$ of $U_t$, we compute one as follows:

   (a) Sample two points $z_1, z_2$ uniformly at random from $U_t$
   (b) Call the ERM oracle with the labeled pair $\{(z_1, 0), (z_2, 1)\}$
   (c) If this labeling is realizable, the oracle returns a concept $c_z \in \mathcal{C}$ with a threshold between $z_1$ and $z_2$. We define:
   - $U_{L,t} = \{x \in U_t : c_z(x) = 0\}$ (points labeled 0 by concept $c_z$)
   - $U_{R,t} = \{x \in U_t : c_z(x) = 1\}$ (points labeled 1 by concept $c_z$)
   (d) If $\min(|U_{L,t}|, |U_{R,t}|) < \frac{1}{3}|U_t|$ (i.e., the partition is too imbalanced), we repeat sampling and partitioning
   (e) By random sampling properties, after $O(\log(1/\delta))$ attempts, we find a balanced partition with probability at least $1 - \delta$

3. **Prediction using Partition:** Once we have a valid partition $(U_{L,t}, U_{R,t})$:

   - If $x_t \in U_{L,t}$, predict $\hat{y}_t = 0$
   - If $x_t \in U_{R,t}$, predict $\hat{y}_t = 1$

4. **Update Rule:** After receiving the true label $y_t$:

   - If our prediction was correct, we maintain the current partitioning
   - If we predicted $\hat{y}_t = 0$ for $x_t \in U_{L,t}$ but received $y_t = 1$ (a mistake), then:
     - All points in $U_{R,t}$ must have label 1
     - Update: $R_{t+1} = R_t \cup U_{R,t}$
     - Update: $L_{t+1} = L_t$
     - Update: $U_{t+1} = U_{L,t}$
     - Invalidate the current partition for the next time step
   - If we predicted $\hat{y}_t = 1$ for $x_t \in U_{R,t}$ but received $y_t = 0$ (a mistake), then:
     - All points in $U_{L,t}$ must have label 0
     - Update: $L_{t+1} = L_t \cup U_{L,t}$
     - Update: $R_{t+1} = R_t$

- Update: $U_{t+1} = U_{R,t}$
- Invalidate the current partition for the next time step

For mistake analysis, each mistake reduces the size of the unknown region $U_t$ by a factor of at least $2/3$, since we ensure that $\min(|U_{L,t}|, |U_{R,t}|) \geq \frac{1}{3}|U_t|$ and at least one of these regions is moved to either $L_{t+1}$ or $R_{t+1}$ after a mistake. Since $|U_1| = T$ and each mistake reduces $|U_t|$ by a factor of at least $2/3$, after $O(\log_{3/2} T) = O(\log T)$ mistakes, we have $|U_t| \leq 5$, at which point we can determine the exact threshold with a constant number of additional mistakes.

For oracle complexity, we make $O(\log(1/\delta))$ ERM oracle calls per mistake to find a balanced partition, where $\delta$ is a small constant. With $O(\log T)$ mistakes, this results in a total of $O(\log T \cdot \log(1/\delta)) = O(\log T)$ ERM oracle calls in expectation.

Therefore, our randomized algorithm makes $O(\log T)$ mistakes and requires $O(\log T)$ ERM oracle calls in expectation. ∎

## H.2 Intervals

We now go into the results for $k$-Intervals ($\mathcal{F}_{int,k}$)

**Proposition H.1 (VC Dimension of $k$-Intervals)** *Given an instance space $\mathcal{X}$ with $|\mathcal{X}| \geq 2k + 1$, for any $\mathcal{C} \in \mathcal{F}$, the VC Dimension of $\mathcal{C}$ is equal to $2k + 1$.*

**Proof** For the lower bound of $2k$, consider any $2k$ points $z_1 \prec z_2 \prec \ldots \prec z_{2k}$ under the ordering $\preceq$. Any labeling of these points is realizable because it requires at most $k$ intervals to separate regions labeled 1, as each maximal contiguous sequence of 1s forms one interval.

For the upper bound, observe that any set of $2k + 1$ points cannot be shattered. Specifically, for points $z_1 \prec z_2 \prec \ldots \prec z_{2k+1}$, the alternating labeling $c(z_i) = i \mod 2$ would require $k + 1$ intervals to realize, which exceeds the capacity of the class. ∎

**Theorem 4.6 (Upper Bound on Transductive Online Learning of $k$-Intervals with Weak Consistency Oracle - Randomized Algorithm)** *Consider transductive online learning with the family $\mathcal{F}_{int,k}$. There exists a randomized algorithm that makes $O(T^3 \cdot 2^{2k})$ calls to the weak consistency oracle in expectation and makes at most $O(k \log T)$ mistakes.*

Our approach to proving this theorem consists of two phases: (1) determining the unknown ordering $\preceq$ that defines the concept class, and (2) applying standard online learning algorithms once the concept class is identified. The key insight is that we can efficiently identify the relative ordering of points by testing which points are at the extremes of the ordering.

**Lemma H.2 (Testing for extreme points)** *Let $\mathcal{C} \in \mathcal{F}_{int,k}$ be an unknown concept class, and consider a set $U \subset \mathcal{X}$ with $|U| \geq 2k + 2$. For any $z \in U$, it requires $O(|U| \cdot 2^{2k})$ weak consistency oracle queries to determine, with high probability, whether $z$ is one of the two endpoints (minimum or maximum) in $U$ with respect to the ordering $\preceq$ corresponding to $\mathcal{C}$.*

To prove the lemma, we also make use of the following lemma:

**Lemma H.3 (Lower bound on mis-labeling probability for non-endpoints)** *Let $z \in U$ be a point that is not an endpoint in the ordering $\preceq$ over $U$, where $|U| \geq 2k + 2$. When $2k$ points are sampled uniformly at random from $U \setminus \{z\}$ without replacement and construct the unique unrealizable alternating labeling over the resulting set of $2k + 1$ points, the probability that $z$ receives label $0$ is at least $\frac{1}{|U|}$.*

**Proof of Lemma H.3** Let $n := |U|$ and $s := 2k$. Since $z$ is not an endpoint, its rank in the total ordering on $U$ is some $r \in \{2, \ldots, n-1\}$. This means there are $A := r - 1 \geq 1$ points below $z$ and $B := n - r \geq 1$ points above $z$. Since both endpoints in the ordering are in the odd position, let's assume without loss of generality that $r \leq \frac{n+1}{2}$, i.e. $r$ is on the "left half" of the ordering.

After sampling $s$ points without replacement from $U \setminus \{z\}$, let $L$ denote the number of sampled points that lie below $z$ in the ordering. Then $L$ follows a hypergeometric distribution:

$$\mathbb{P}[L = \ell] = \frac{\binom{A}{\ell}\binom{B}{s-\ell}}{\binom{n-1}{s}}.$$

In the alternating labeling pattern of the $2k + 1$ points, $z$ receives label 0 if and only if it occupies an even position in the sorted order, which occurs precisely when $L$ is odd.

Define $P_{\text{even}} = \sum_{\ell \text{ odd}} \mathbb{P}[L = \ell]$ and $P_{\text{odd}} = \sum_{\ell \text{ even}} \mathbb{P}[L = \ell]$. By the alternating Vandermonde identity [JSJ+15], we have:

$$P_{\text{even}} - P_{\text{odd}} = \frac{(-1)^s \binom{B-A-1}{s}}{\binom{n-1}{s}}.$$

Since $A, B \geq 1$, we have $|B - A - 1| \leq n - 3$, which gives us:

$$|P_{\text{even}} - P_{\text{odd}}| \leq \frac{\binom{n-3}{s}}{\binom{n-1}{s}} = \frac{(n-1-s)(n-2-s)}{(n-1)(n-2)} \leq 1 - \frac{s}{n-1}.$$

Therefore:

$$P_{\text{even}} = \frac{1}{2}(1 + (P_{\text{even}} - P_{\text{odd}})) \geq \frac{1}{2}(1 - |P_{\text{even}} - P_{\text{odd}}|)$$

$$\geq \frac{1}{2}\left(1 - \left(1 - \frac{s}{n-1}\right)\right) = \frac{s}{2(n-1)} = \frac{k}{n-1}.$$

Since $n = |U| \geq 2k + 2$ and $k \geq 1$, we conclude that $P_{\text{even}} \geq \frac{k}{|U|-1} \geq \frac{1}{|U|}$. $\blacksquare$

We prove Lemma H.2 below.

**Proof of Lemma H.2** We first establish a critical property of $k$-interval concept classes: For any sequence of $2k + 1$ points $z_1 \prec z_2 \prec \ldots \prec z_{2k+1}$ arranged according to the underlying ordering $\preceq$, the alternating labeling $c(z_i) = i \mod 2$ is the unique labeling that cannot be realized by any concept in the class. This is because any $k$-interval concept can create at most $k$ label transitions when moving from one point to the next in the ordered sequence, but the alternating labeling requires $2k$ transitions.

Now, consider performing the following process:

1. Randomly sample $2k$ points from $U \setminus \{z\}$ without replacement.

2. Consider the set $S$ consisting of these $2k$ sampled points together with $z$.

3. Using the weak consistency oracle, test all $2^{2k+1}$ possible labelings of $S$ to identify the unique unrealizable labeling.

4. Observe the label assigned to $z$ in this unrealizable labeling.

The key insight is that in the unique unrealizable labeling (the alternating pattern), the position of $z$ in the ordering determines its label:

- If $z$ is the minimum element in $S$ (i.e., $z \prec z'$ for all $z' \in S \setminus \{z\}$), it will be labeled 1 in the unrealizable labeling.

- If $z$ is the maximum element in $S$ (i.e., $z' \prec z$ for all $z' \in S \setminus \{z\}$), it will be labeled 1 in the unrealizable labeling.

- If $z$ is neither the minimum nor maximum in $S$, its label in the unrealizable labeling depends on its position in the sequence and will be either 0 or 1, depending on whether the position is odd or even.

We now analyze the probability of correctly identifying whether $z$ is an endpoint:

- If $z$ is truly an endpoint (minimum or maximum) in $U$, then in every sample $S$, $z$ will be at the same extreme position in the ordering of $S$. Consequently, $z$ will receive the same label (either consistently 1 for the minimum, or consistently 1 for the maximum since $|S| = 2k+1$ is odd) in the unrealizable labeling across all samples.

- If $z$ is not an endpoint in $U$, then in different random samples, the probability that $z$ is labeled as 0 in a realizable labeling, by Lemma H.3, is at least $\frac{1}{|U|}$.

To ensure correctness with high probability (at least $1 - \delta$ for some small constant $\delta$), we repeat this sampling process $O(|U| \cdot \log \frac{1}{\delta})$ times. By the properties of independent trials, if $z$ is not an endpoint, we will observe a contradictory label with high probability after these repetitions.

The total number of weak consistency oracle queries required is:

$$O(|U| \cdot \log \frac{1}{\delta} \cdot 2^{2k+1}) = O(|U| \cdot 2^{2k}) \tag{1}$$

when setting $\delta$ to be a small constant. ∎

Now we prove the theorem.

**Proof of Theorem 4.6** Our approach consists of two phases: (1) determining the unknown ordering $\preceq$ that defines the concept class with high probability, and (2) applying a standard online learning algorithm once the concept class is identified.

**Phase 1: Determining the ordering.** We will identify the ordering of the input points $\{x_1, \ldots, x_T\}$ by successively identifying extreme points:

1. Initialize $U = \{x_1, \ldots, x_T\}$ and an empty sequence $S$.

2. While $|U| > 2k + 1$:

   (a) Identify extreme points $z_1, z_2$ using Lemma H.2 by iterating over all $T$ points.
      - If it's the first iteration (i.e. $|U| = T$), designate $z_2$ as $z^*$. $z^*$ will be the same fixed point for the remaining iterations.
      - For whichever $i \in \{1, 2\}$ such that $z_i \neq z^*$, append $z$ to $S$ and remove it from $U$.

3. The remaining points in $U$ (at most $2k + 1$) can be placed in any order at the end of $S$.

This procedure yields a sequence $S = (z_1, z_2, \ldots, z_{T-|U|}, \ldots)$ which, with high probability, represents either the correct ordering $\preceq$ or its reverse. Since the definition of the interval concept class is symmetric with respect to ordering direction, we can arbitrarily choose one direction.

To analyze the number of queries, when identifying the extreme points $z_1$ and $z_2$ using Lemma H.2, at most $T$ points are tested, which involves $O(T^2 2^{2k})$ queries. There are $T$ iterations in the algorithm, giving that there are $O(T^3 2^{2k})$ queries.

**Phase 2: Learning with the determined ordering.** Once we have determined the ordering $\preceq$ with high probability, we know that the target concept belongs to the class $\mathcal{C}_{\text{int},\preceq,k}$. This class has VC dimension $2k$ and Littlestone dimension $O(k \log T)$. We thus know the concept class for all but at most $2k$ points. We can now apply the Standard Optimal Algorithm (SOA) or the halving algorithm for this concept class. When seeing one of the $2k$ points for where the ordering is uncertain, predict arbitrarily. These algorithms guarantee a mistake bound of $O(k \log T)$ in the realizable case, and the $2k$ uncertain points contribute at most $2k$ points, resulting in an $O(k \log T)$ mistake bound. ∎

## H.3 Hamming Balls

For a set $\mathcal{X}$, we define the family of concept classes $\mathcal{F}_d = \{\mathcal{C}_{f,d} \mid f : \mathcal{X} \to \{0, 1\}\}$, where each $\mathcal{C}_{f,d}$ represents the set of concepts that differ from $f$ on at most $d$ points:

$$\mathcal{C}_{f,d} = \{g : \mathcal{X} \to \{0,1\} \mid |\{x \in \mathcal{X} : g(x) \neq f(x)\}| \leq d\}$$

These classes have several important properties. First, each $\mathcal{C}_{f,d}$ has Littlestone dimension exactly $d$. To establish the lower bound, we can select any $d$ distinct points $x_1, x_2, \ldots, x_d \in \mathcal{X}$ and construct a Littlestone tree of depth $d$ by placing $x_i$ at level $i$. By definition of $\mathcal{C}_{f,d}$, all $2^d$ possible labelings of these points are achievable by some concept in the class, thus confirming that the Littlestone dimension is at least $d$.

For the upper bound, suppose for contradiction that there exists a Littlestone tree of depth $d + 1$ that is shattered by $\mathcal{C}_{f,d}$. Let $z_1$ be the root node of this tree. For each $i \in \{1, 2, \ldots, d\}$, define $z_{i+1}$ recursively as follows: if $f(z_i) = 0$, let $z_{i+1}$ be the right child of $z_i$; otherwise, let $z_{i+1}$ be the left child of $z_i$. This construction yields a concept $c \in \mathcal{C}_{f,d}$ that must satisfy $c(z_i) \neq f(z_i)$ for all $i \in \{1, 2, \ldots, d + 1\}$. But this implies that $c$ disagrees with $f$ on $d + 1$ points, contradicting the definition of $\mathcal{C}_{f,d}$. Therefore, the Littlestone dimension is exactly $d$.

**Theorem H.4 (Transductive Online Learning Bounds for $d$-Hamming Balls)** *Let $\mathcal{F}_d$ be the family of $d$-Hamming Balls as defined above. Then:*

1. *There exists a deterministic algorithm that makes $O(1)$ calls to the ERM oracle and incurs at most $O(d)$ mistakes.*

2. *There exists a deterministic algorithm that makes $O(T)$ calls to the weak consistency oracle and incurs at most $O(d)$ mistakes.*

Before proving the theorem, we comment that in the transductive online learning setting, one can make a reduction from the weak consistency oracle to ERM, achieving the same mistake bound, but with an additional $O(T)$ factor. This comes from the observation that ERM solves a search problem (it searches for a concept), while the weak consistency oracle solves a decision problem (it determines whether or not a dataset is realizable), and ERM to return a function restricted to $T$ points can be solved by running our decision problem $T$ times. This is captured via the following lemma.

**Lemma H.5 (Reducing ERM Oracle to Weak Consistency Oracle)** *Consider a transductive online learning algorithm that makes at most $Q$ ERM oracle calls where each call only uses points from the set $\{x_1, \ldots, x_T\}$ as inputs, only evaluates the returned concepts on these points, and makes at most $M$ mistakes. Then there exists an algorithm that uses at most $TQ$ weak consistency oracle calls and makes at most $M$ mistakes.*

**Proof** We simulate each ERM call using at most $T$ weak consistency calls. For each ERM query on a set $S = \{(x_{i_1}, y_{i_1}), \ldots, (x_{i_k}, y_{i_k})\}$ with $k \leq T$ and all $x_{i_j} \in \{x_1, \ldots, x_T\}$, we construct a concept $c$ as follows:

First, we check if $S$ is realizable by making a weak consistency call on $S$. If $S$ is not realizable, we return "not realizable" as the ERM oracle would.

If $S$ is realizable, we construct a concept $c \in \mathcal{C}$ consistent with $S$ by determining the values of $c$ on each point in $\{x_1, \ldots, x_T\}$:

- For points already in $S$, we set $c(x_{i_j}) = y_{i_j}$ for all $j \in [k]$.

- For each $x_j \in \{x_1, \ldots, x_T\} \setminus \{x_{i_1}, \ldots, x_{i_k}\}$, we query the weak consistency oracle with $S \cup \{(x_j, 1)\}$. If the result is true, we set $c(x_j) = 1$; otherwise, we set $c(x_j) = 0$ (as there must exist a concept in $\mathcal{C}$ consistent with $S$ that assigns 0 to $x_j$).

This procedure uses at most $T$ weak consistency calls per ERM query (one initial call to check realizability, and at most $T - k$ additional calls for the remaining points). Since the original algorithm makes at most $Q$ ERM calls, the total number of weak consistency calls is at most $TQ$.

Furthermore, since we accurately simulate each ERM call by returning a concept consistent with the query set whenever such a concept exists, and the original algorithm only evaluates the returned concepts on points in $\{x_1, \ldots, x_T\}$, the mistake bound $M$ of the original algorithm is preserved. ∎

Now we prove the theorem.

**Proof of Theorem H.4** For the ERM oracle bound, the algorithm begins by querying the oracle with the empty set to obtain a function $c \in \mathcal{C}_{f,d}$. By definition, both $c$ and the target concept are at most $d$ away from the center function $f$, so by the triangle inequality, they can disagree on at most $2d$ points. The algorithm simply uses $c$ to make all its predictions, resulting in at most $2d$ mistakes with just a single ERM oracle query.

For the weak consistency oracle bound, we apply Lemma H.5. Since our ERM-based algorithm only evaluates the returned concept on the points $x_1, \ldots, x_T$ and makes $O(1)$ ERM calls, we can simulate it using $O(T)$ weak consistency oracle calls while maintaining the same mistake bound of $O(d)$. ∎

Furthermore, it is possible to get the optimal transductive mistake bound, if one relaxes the number of queries needed.

**Theorem H.6 ($d$-Hamming Balls, Optimal Mistake Bound)** *Consider transductive online learning with $\mathcal{F}_d$, the family of $d$-Hamming balls. Then, optimal mistake bounds can be achieved using at most $2^{d+1}$ queries using the ERM oracle.*

**Proof** Consider $x_1, x_2, \ldots, x_{d+1}$, and let the unknown concept class be $\mathcal{C}_{f,d}$ There is exactly one non-realizable labeling of these $d+1$ points (since every other labeling disagrees with $f$ by at most $d$ points). Then, one can recover $f(x_1), f(x_2), \ldots, f(x_{d+1})$ via flipping the labels in the realizable labeling. Perform an ERM query with inputs $\{(x_1, 1 - f(x_1)), \ldots (x_d, 1 - f(x_d))\}$. The oracle will output the value of $f$ for $x_{d+1}, \ldots, x_T$, allowing the learner to gain knowledge of the concept class, from which they can perform standard transductive online learning. ∎

