# OpenReview forum: "Tradeoffs between Mistakes and ERM Oracle Calls in Online and Transductive Online Learning"
_NeurIPS.cc/2025/Conference — NeurIPS 2025 spotlight_

### Official Review · Reviewer_SSWL · 2025-06-27

**Clarity:** 4
**Significance:** 3
**Originality:** 3
**Rating:** 5
**Confidence:** 4

**Summary:**

The paper continues the study of online learning (and transductive online learning) in restricted access models. While the problem of online is very well understood, most results are based on algorithms that require the knowledge of the hypothesis class and perform some intractable computations. This paper considers restricted setting where the algorithm has either access to an ERM oracle, a restricted ERM oracle that can only query the ERM on points seen so far, and a weak consistency oracle that only identifies if the sample is realizable or not.

The authors prove several interesting results in these settings. One of the results shows a strong exponential in terms of the Littleston dimesnion lower bound on the mistake of any learner that uses ERM oracles, which matches the known upper bound. Another interesting result is the connection between weak consistency oracles and restricted ERMs. They show that any algorithm that uses restricted ERMs can be simulated by an algorithm that uses weak consistency oracle with a multiplicative O(T) factor in the query complexity.

In the transductive setting, a significant result is showing that with $O(T^{d_{VC}}$ many oracle calls, it is possible to obtain the optimal mistake bound even in the weak consistency oracle setting. It is further showed that $O(T)$ oracle calls is not sufficient even for the ERM oracle. Moreover, mistake and oracle bounds are derived for specific classes of interest.

**Questions:**

Is it clear whether the doubly exponential calls for ERM oracle is necessary and is the oracle calls for randomized algorithms of the same order?

Is it clear if the multiplicative $O(T)$ factor in oracle calls for reducing weak consistency model to restricted ERM model necessary? Moreover, it makes sense from the lower bound in the transductive setting that a lower bound for restricted ERM will imply a lower bound for the weak consistency model too. But in the online learning, there are only lower bounds for the restricted ERM model. I think that this should also imply some lower bounds for consistency model but it is not completely clear to me. The reason is that the weak consistency oracle can query the realizability of samples that are not already observed too.

**Ethical Concerns:**

["NO or VERY MINOR ethics concerns only"]

**Final Justification:**

My initial assessment of the paper was positive. After reading rebuttals, I still keep my score. The authors explained well my questions and I agree that some of the questions are for future work. I raised them because the paper provokes them, which makes me believe the problem studied is an interesting research direction.

**Limitations:**

Yes.

**Paper Formatting Concerns:**

None.

**Quality:**

4

**Strengths And Weaknesses:**

I guess the question of what is achievable in models where the computation is more restricted is an important question in learning theory and the authors seem to make significant steps in the online learning direction. I also believe showing that learning in such restricted models with respect to partial concepts is impossible is an interesting result, which makes point to the challenge of learning with respect to such complicated models that are hoped to capture to some extent the data assumptions.

I think one important part, which is also mentioned in the paper, is that it is not clear how powerful are randomized algorithms and what mistake bounds is attainable if we are allowed polynomially (in $T$) many oracle calls, independent of VC dim. The only negative result in the paper shows insufficiency of linear number of oracle calls for the weak consistency oracle. If I am not mistaken, it is not even clear what happens to the randomized learners with ERM oracles for polynomially many oracle calls. I think this can be an important future direction for this

---

> ### Author Rebuttal · Authors · 2025-07-31
>
> We thank the reviewer for their knowledgeable review and interesting questions.
>
> Q: "Is it clear whether the doubly exponential calls for ERM oracle is necessary and is the oracle calls for randomized algorithms of the same order?"
>
> A: The picture is still unclear, and whether a merely exponential number of calls suffices remains an interesting open question. The best known upper bound, due to [AAD+23], comes from applying an agnostic‑to‑realizable reduction, which already yields a number of calls exponential in the number of mistakes (itself exponential in their analysis). A more sophisticated algorithm (possibly randomized) might improve this bound, but any such approach would likely be quite involved and far from straightforward.
> On the other hand, it is possible that a matching lower bound will show that the doubly‑exponential dependence is indeed unavoidable.
>
> Q: Is it clear if the multiplicative $O(T)$
>  factor in oracle calls for reducing weak consistency model to restricted ERM model necessary?
>
>
> A: In general, we expect a blowup in the number of oracle calls to the decision problem oracle (weak consistency) compared to a variant of the optimization problem (ERM), as is typically the case in computational settings.
> In the offline (PAC) setting, there is also a polynomial blowup in the number of weak consistency calls compared to a single ERM call (see the paper "Is Efficient PAC Learning Possible with an Oracle That Responds 'Yes' or 'No'?").
> We similarly expect an additional cost in oracle calls in the online model (depending on $T$), though it is possible that a better bound could be achieved, perhaps with only additive or otherwise improved dependence on $T$.
>
> You raise an excellent point about lower bounds. Our restricted ERM lower bounds do not immediately carry over to the weak consistency setting, since weak consistency oracles can query arbitrary datasets, potentially making them more powerful. It is an interesting question whether the weak consistency oracle can ever be strictly stronger than the restricted ERM oracle. All we currently know is that the query complexity of the weak consistency oracle is at least that of the realizable ERM oracle, and at most $T$ times that of the restricted ERM oracle.

---

> > ### Comment · Reviewer_SSWL · 2025-08-09
> >
> > I am happy that you found the questions interesting and thanks for addressing them.

---

### Official Review · Reviewer_1WTD · 2025-07-02

**Clarity:** 3
**Significance:** 3
**Originality:** 3
**Rating:** 4
**Confidence:** 5

**Summary:**

This paper studies online and transductive learning in settings where the learner can only interact with the hypothesis class through ERM or optimization oracles of varying power. Statistically optimal algorithms in the online setting (such as SOA) and the transductive setting typically require direct knowledge of and access to the hypothesis class. This paper asks whether optimal regret/mistake bounds are achievable when the learner can only access the concept class using ERM or other types of optimization and consistency oracles. Foreshadowing that the optimal regret is too much to hope for, the paper aims to understand the tradeoff between the number of oracle calls and the regret/mistake bound broadly.

For the online setting, the paper shows that the mistake bound is lower bounded by $\Omega(2^{LD})$ or $\Omega(3^{LD})$ for randomized and deterministic algorithms, respectively, regardless of the number of oracle calls. In the agnostic case, the regret is lower bounded by $\Omega(\sqrt{T 2^{LD}})$. The authors also provide upper bounds that demonstrate exponential dependence on the Littlestone dimension (LD). For the transductive setting, they show that achieving optimal regret/mistake bounds is possible in $O(T^{VC})$ even using a weaker oracle model. While some bounds improve in this setting, it appears that even in the transductive setting, a no-regret algorithm still requires exponential dependence on LD unless a linear number of oracle calls is allowed.
I find the paper overall interesting. The study of oracle efficiency is of long-standing interest to the COLT community --- which has a good presence within NeurIPS --- and is motivated by real concerns about the practical efficiency of learning algorithms. I find this to be a solid contribution for the theoretically minded NeurIPS audience and I lean towards acceptance.

**Questions:**

See above two points on explicit comparison to Hazan and Koren and motivation given the broader context.

**Ethical Concerns:**

["NO or VERY MINOR ethics concerns only"]

**Final Justification:**

Good paper, as I stated in my review. It addresses an interesting and important problem on oracle efficiency — a core question in the ML theory community — and provides compelling bounds. As discussed in the review, more of the related works should move to the main body, which is currently a bit sparse on the discussion of related works. I recommend acceptance weakly, and given where the scores are, I expect the paper will be accepted.

My only hesitation in recommending acceptance more strongly is that the paper’s message may be limited to the COLT/ALT subcommunity within NeurIPS. I do like that the paper takes a very general and adversarial perspective. But, because we know that many of the known technical impossibilities and challenges in dealing with oracles resolve once we consider models where the adversary is a bit more realistic (e.g., smoothed analysis, as mentioned in my review), I think the paper’s broader appeal beyond the COLT/ALT community may be limited.

**Limitations:**

yes

**Quality:**

3

**Strengths And Weaknesses:**

That being said, I would have felt much more inclined to recommending acceptance more strongly if the paper went the additional mile of 1) connecting with the broader literature a but better, 2) justifying and motivating the study of their work in such generality under such strong adversarial assumptions. Detailed comments below:

I found it difficult to fully connect and compare the findings of this paper to the broader literature on oracle efficiency in online learning, e.g.:
1. Hazan and Koren, The computational power of optimization in online learning, STOC 2016
2. Kalai and Vempala, Efficient algorithms for online decision problems, JCSS 2005
3. Dudík et al., Oracle-efficient Online Learning and Auction Design, FOCS 2017

For example, [1] provides upper and lower bounds on the number of oracle calls required for achieving vanishing regret, though the dependence on $\mathrm{LD}$ or $\mathrm{VC}$ isn't explicitly considered in their work. Are those results subsumed by yours, or do they highlight tradeoffs that are qualitatively different? The techniques shown in [2–3] do not apply to general VC or LD classes (as [1] already establishes), but given the relevance of your work to the oracle-efficient online learning literature, I would have expected a discussion of all of these works.

The paper would also benefit from a clearer motivation for studying this particular model at the generality of VC or LD classes. I appreciated the improved bounds for certain classes in the transductive setting, but my concern is not a lack of results, but rather a lack of context for paper broadly. There are some minimal attempts at this (e.g., lines 56 on computability issues) but I find the discussion severely lacking.

Since this paper is being submitted to NeurIPS and not COLT, the authors should embrace the additional responsibility of communicating to a broader audience beyond the TCS community.

This is especially important given that in both theory and practice, a number of works have demonstrated that oracles are quite powerful in many practical online learning settings. For example, [2,3] demonstrate that oracle efficiency and no-regret guarantees can be achieved in settings that are broadly of interest to the community. Moreover, for general VC classes, oracles have also been shown to be quite powerful outside brittle counter examples. For instance:

4. Haghtalab et al., Oracle-Efficient Online Learning for Beyond Worst-Case Adversaries, NeurIPS 2022
5. Block et al., On the Performance of Empirical Risk Minimization with Smoothed Data, COLT 2024

In particular, [5] shows that repeated “restricted ERM” suffices for achieving no-regret in realizable online learning if the marginal distribution is smooth. Similarly, [4] studies the agnostic setting under smoothness and also shows that ERM oracle calls suffice for online learning of VC classes. This smoothness condition has come to be widely accepted by the community as a useful test for whether impossibility results against adversaries hold beyond brittle counter examples and demonstrate practical concerns.  Looking at your lower bounds, the constructions also demonstrate quite a lot of brittleness because they are based on requiring exact identification of randomly chosen real values which cannot be done with finite number of oracle calls.

In light of these results—and the broader NeurIPS audience—I would have liked to see a more rigorous discussion of why the fully adversarial setting in your model is worth studying and why should it be interesting to the broader NeurIPS community. What should be the takeaways from reading your paper in terms of practical tradeoffs in online environments?


Minor:
•	Line 190: “a a concept” → “a concept”

---

> ### Author Rebuttal · Authors · 2025-07-30
>
> We thank the reviewer for their thoughtful review and constructive feedback, and we are happy to implement the suggestions for improving our presentation.
>
> 1. Connecting with broader literature:
> We already discuss most of the papers you mentioned in the "Additional Related Work" section (specifically, the two paragraphs in the Oracle-Efficient Online Learning part of the appendix). However, we will expand on this discussion in more detail and move it into the main body of the paper to provide better context for our contributions. See the new related work section on oracle-efficient online learning at the end of our response.
>
> 2. "Justifying and motivating the study of such generality under strong adversarial assumptions":
> - Our setting is both highly generic and strongly adversarial, which we view as a strength. Studying learning under such restrictive conditions helps us understand the fundamental barriers, so that any additional assumptions introduced later are clearly motivated and justified. We adopt only the minimal assumptions necessary for learning in this setting: that the (possibly infinite) concept classes have bounded combinatorial dimension, a standard requirement in general online learning, and that the instance sequence is adversarial.
> Our approach is aligned with prior work [HK16, AAD+23, KS24], and it resolves open questions under this minimal set of assumptions.
> - By illuminating this general setting, our work opens the door to studying richer regimes. In particular, we view it as a natural next step to explore “beyond worst-case” models by introducing assumptions on the data-generating process or limiting the adversary’s power, as you suggest. For instance, the smoothed analysis framework proposed by Spielman and Teng was developed only after decades of studying the Simplex algorithm (and other linear programming methods) in their most general form.
> We will add a paragraph to the discussion section proposing future directions that incorporate such assumptions, and we will cite the works by Haghtalab et al. (2022) and Block et al. (2024). We will also include a discussion of these papers in the revised related work section.
>
> We also thank you for catching the typo on line 190, and we will fix it accordingly.
>
> We are happy to answer any further questions and hope our response supports your strong recommendation for acceptance
>
> - Below are the paragraphs that we plan to add to the related work in the "Oracle-Efficient Online Learning" section. Specifically, we expand the second paragraph with the following:
>
> Theoretical limitations of this approach have also been investigated. Hazan and Koren [HK16] showed that, for certain finite concept classes $\mathcal{C}$, any oracle-efficient online learner must make at least $\tilde{\Omega}(\sqrt{|\mathcal{C}|})$ calls to a powerful optimization oracle (an agnostic ERM oracle) to achieve sublinear regret in the fully adversarial proper-learning setting; they also give a matching $\widetilde{O}(\sqrt{|\mathcal{C}|})$ upper bound. Since the Littlestone dimension satisfies $\operatorname{Ldim}(\mathcal{C}) \le \log_2 |\mathcal{C}|$ for finite classes, this implies an exponential dependence of total runtime on the Littlestone dimension. Our results sharpen and generalize this picture: we give lower bounds in \emph{both} the realizable and agnostic regimes, and for general (possibly infinite) classes parameterized by VC / Littlestone dimension. Rather than targeting mere sublinear regret, we identify conditions under which stronger, dimension-dependent rates are unattainable for any oracle-efficient learner—e.g., $\Omega(2^{d_{\mathrm{LD}}})$ (realizable) and $\Omega\big(\sqrt{T\,2^{d_{\mathrm{LD}}}}\big)$ (agnostic). When specialized to finite classes with $|\mathcal{C}| \approx 2^{d_{\mathrm{LD}}}$, both lines of work exhibit exponential dependence on $d_{\mathrm{LD}}$ (since $\sqrt{|\mathcal{C}|} = 2^{d_{\mathrm{LD}}/2}$), while our results additionally showed that even in the realizable case the required computation is exponential in $d_{\mathrm{LD}}$ (and in fact cannot be done in a finite number of queries). Altogether, these findings underscore that optimization oracles remove optimization difficulty but not the information-theoretic hardness of fully adversarial online learning.
>
> Additionally, Kalai and Vempala [KV05] and Dudík et al. [DHL+20] developed oracle-efficient algorithms for various online learning problems (e.g., combinatorial decisions and auctions) by leveraging structure or randomness (perturbed-leader methods), but their techniques assume an efficient oracle for the specific problem at hand and do not apply to arbitrary concept classes.
>
> Several works have identified structural conditions that enable oracle-efficient online learning despite the general lower bounds. Dudík et al. [DHL+20] studied the conditions under which oracle-efficient algorithms can succeed, and Haghtalab et al. [HHSY22] provided such algorithms for online learning with smoothed adversaries, introduced in Haghtalab et al. [HRS24]. Haghtalab et al. (2022) gave the first oracle-efficient online learner in the smoothed adversary model, achieving regret bounds $O(\sqrt{T,d/\sigma})$ that depend only on the hypothesis class’s VC dimension $d$ and the smoothness parameter $\sigma$. Their result shows that when the adversary is constrained to draw instances from a $\sigma$-smooth distribution (i.e., no individual example can have too large a probability mass), online learning becomes computationally as easy as offline learning for any VC class. More recently, Block et al. [BRS24] considered the realizable case under smooth adversaries and show that even a simple repeated-ERM strategy can attain no-regret. In particular, they proved that if the marginal distribution is $\sigma$-smooth, then empirical risk minimization achieves sublinear error on the order of $\tilde O(\sqrt{\mathrm{comp}(\mathcal{F}) \cdot T})$, where $\mathrm{comp}(\mathcal{F})$ is the standard PAC-learning complexity of the class. These works underscore that smoothness assumptions, now widely used as a testbed for the robustness of impossibility results – can circumvent the brittle worst-case constructions, enabling efficient online learning of VC classes in practice. However, studying the models for general VC / Littlestone classes is necessary for gaining more fundamental insights into the computational-statistical tradeoffs that govern all concept classes. Studying general classes reveals which properties are universal versus which depend on special structure. While [HHSY22] and [HRS24] establish oracle-efficient algorithms under smoothness assumptions, oracle efficiency in the fully adversarial setting has yet to be resolved.

---

> > ### Comment · Reviewer_1WTD · 2025-08-03
> >
> > I thank the authors for the expanded related work discussion. I agree that this (and as much as possible more of what's currently in the appendix) should be included in the main body. At the moment, the main body is quite sparse on related works and broader connections/implications.
> >
> > On the second point, I agree about using worst-case analysis as guidance for where and how to go beyond the worst case. The community has made strides on the beyond-the-worst-case front on the problem you are considering, but having a clearer picture of the worst-case tradeoffs provides more ammunition to push forward in those directions still! My feedback here was more about situating your work in a way that clarifies whether any and what the kind of impact you hope to see on the broader NeurIPS community, outside of the COLT/ALT circle. What you have committed to in terms of revising your related works section does that to a degree. Beyond that, I am happy to accept that not every ML theory work appearing at NeurIPS needs to have a broader message or be practically relevant. NeurIPS has a vibrant community of ML theory researchers who will appreciate your results.
> >
> > I will keep my score as is for now and I'm happy to that the paper's merits are appreciated by other reviewers as well.

---

### Official Review · Reviewer_8aS9 · 2025-07-06

**Clarity:** 4
**Significance:** 3
**Originality:** 3
**Rating:** 5
**Confidence:** 3

**Summary:**

This paper investigates online and transductive online learning scenarios where the learner interacts with the concept class solely through ERM or weak consistency oracles. The authors establish tight lower bounds for the number of mistakes and oracle calls in both realizable and agnostic settings, demonstrating an exponential dependence on the Littlestone dimension. They also explore the transductive setting, where instance sequences are known in advance, and show that polynomial oracle calls can achieve optimal mistake bounds. Special cases like thresholds and k-intervals are analyzed, revealing improved performance with randomized algorithms.

**Questions:**

The transductive setup is a good bypass for the hardness of online learning. Out of curiosity, have you considered other routes to bypass, e.g., i.i.d. assumptions on the covariates, margin assumptions for linear separators, and smoothed online learning?

**Ethical Concerns:**

["NO or VERY MINOR ethics concerns only"]

**Quality:**

4

**Strengths And Weaknesses:**

Strengths:
1. The paper provides rigorous lower and upper bounds, clearly delineating the trade-offs between mistakes and oracle calls. The proofs are well-structured and detailed, with key insights highlighted.
2. The exploration of weak consistency oracles in online learning is novel, bridging gaps between PAC learning and online settings. The results for specific concept classes (e.g., thresholds, k-intervals) offer insights.

Weaknesses:
No specific one.

---

> ### Author Rebuttal · Authors · 2025-07-31
>
> We thank the reviewer for their thoughtful review and positive assessment of our work.
>
> Other routes to bypass the fully adversarial online setting:
> This is a great suggestion, and we agree that it points to a natural and important direction for future work. In our revision, we will expand the discussion to include potential relaxations of the model, such as making assumptions on the data-generating process or limiting the power of the adversary. For example, one could assume structure on the covariates or consider a smoothed analysis setting as you suggested.

---

### Official Review · Reviewer_uFrM · 2025-07-17

**Clarity:** 4
**Significance:** 3
**Originality:** 4
**Rating:** 5
**Confidence:** 1

**Summary:**

This paper studies the number of mistakes and regret achievable when a learner can only interact with the concept class via Empirical Risk Minimization Oracle or weak consistency oracles.
Unlike standard online models where the learner has full knowledge of the concept class, as here, the learner interacts with the concept class solely through various oracles. The main motivation comes from the computational intractability of implementing algorithms like the Standard Optimal Algorithm (SOA), which, despite being optimal, involves computing Littlestone dimension multiple times even though it is APX-hard. As a result they work with an ERM oracle which given a labeled dataset returns a concept class which minimizes the error on the dataset.  This oracle is known to be sufficient in PAC learning settings. They also consider a weak-consistancy oracle which only returns a binary signal whether the dataset is realizable by some concept in the class.

Their results are as follows:
A $\Omega(2^{d_L})$ mistake lower bound in the realizable setting. They also show a $\Omega\sqrt{T 2^{d_L}})$ on the regret in teh agnostic setting. Their result establishes that an exponential dependence on the little-stone dimension is essential.  The proof  for this lower bound involves constructing a concept class of random threshold functions over $T = 2^{d_L}$ points embedded in a high-dimensional space. The adversary carefully defines nested hyper-rectangles using random values that are unknown to the learner. Since the (agnostic) ERM oracle does not know the future nested regions  the learner is forced to make predictions with only 1/2 probability of being correct at each step.

They also show that any deterministic online learning algorithm that uses the restricted ERM oracle can be simulated using the weaker consistency oracle. The cost is an increase in the number of oracle calls by a factor of O(T), while maintaining the same mistake bounds. This implies that existing ERM-based algorithms can be adapted for the weak consistency setting.

They also show a number of results for transductive learning, which effectively shows that the negative results in the online setting can be circumvented here. They also show several new results for specific concept classes (all of which are  summarized in Table 2). Unfortunately, I am not as familiar with this part of the literature and am not sure I can judge their novelty.

**Questions:**

N/A

**Ethical Concerns:**

["NO or VERY MINOR ethics concerns only"]

**Final Justification:**

I am happy with the response the authors provide regarding the novelty of their work. I am happy to keep my review as is.

**Quality:**

3

**Strengths And Weaknesses:**

I feel the authors establish an important lowerbound for ERM oracles in online learning. This is in line with recent works that have tried to understand better algorithms and oracles that circumvent known hardness results. I found the paper to be well written and the results clear. The proofs are mostly in the appendix but they seemed correct.

I am not an expert in the area and have little background knowledge. Nonetheless, I think the paper would be of interest to the NeurIPS community.

---

> ### Author Rebuttal · Authors · 2025-07-31
>
> We thank the reviewer for their positive assessment of our paper.

---

### Decision · Program_Chairs · 2025-09-17

**Decision:**

Accept (spotlight)

**Comment:**

This paper presents a solid theoretical contribution to the study of online and transductive learning,. The authors establish tight lower bounds on mistakes and regret in the online setting, showing that even with powerful oracles, the performance is fundamentally limited by the Littlestone dimension. All reviewers agreed that it is a strong candidate for acceptance.